# NOTCH1 activation compensates BRCA1 deficiency and promotes triple-negative breast cancer formation

Kai Miao [1,2], Josh Haipeng Lei[1,2], Monica Vishnu Valecha[1,2], Aiping Zhang[1,2], Jun Xu[1,2], Lijian Wang[1,2], Xueying Lyu[1,2], Si Chen[1,2], Zhengqiang Miao[1,3], Xin Zhang[1,4], Sek Man Su[1,2], Fangyuan Shao[1,2], Barani Kumar Rajendran[1,2], Jiaolin Bao[1,2], Jianming Zeng [1,2], Heng Sun[1,2], Ping Chen[1,2], Kaeling Tan[1,3], Qiang Chen[1,2], Koon Ho Wong [1,3], Xiaoling Xu [1,2,4] & Chu-Xia Deng [1,2✉]

BRCA1 mutation carriers have a higher risk of developing triple-negative breast cancer (TNBC), which is a refractory disease due to its non-responsiveness to current clinical targeted therapies. Using the Sleeping Beauty transposon system in Brca1-deficient mice, we identified 169 putative cancer drivers, among which *Notch1* is a top candidate for accelerating TNBC by promoting the epithelial-mesenchymal transition (EMT) and regulating the cell cycle. Activation of NOTCH1 suppresses mitotic catastrophe caused by BRCA1 deficiency by restoring S/G2 and G2/M cell cycle checkpoints, which may through activation of ATR-CHK1 signalling pathway. Consistently, analysis of human breast cancer tissue demonstrates NOTCH1 is highly expressed in TNBCs, and the activated form of NOTCH1 correlates positively with increased phosphorylation of ATR. Additionally, we demonstrate that inhibition of the NOTCH1-ATR-CHK1 cascade together with cisplatin synergistically kills TNBC by targeting the cell cycle checkpoint, DNA damage and EMT, providing a potent clinical option for this fatal disease.

[1] Cancer Center, Faculty of Health Sciences, University of Macau, Macau, SAR, China. [2] Centre for Precision Medicine Research and Training, Faculty of Health Sciences, University of Macau, Macau, SAR, China. [3] Genomics & Bioinformatics Core, Faculty of Health Sciences, University of Macau, Macau, SAR, China. [4] Transgenic and Knockout Core, Faculty of Health Sciences, University of Macau, Macau, SAR, China. ✉email: cxdeng@umac.mo

B reast cancer gene 1 (BRCA1), is the first identified breast cancer susceptibility gene, and is responsible for ~20–25% of hereditary breast cancers and 5–10% of total breast cancers[1]. Low levels of BRCA1 expression are detected in ~30% of sporadic breast cancers, possibly due to hypermethylation of the promoter or other transcriptional regulatory mechanisms[2,3]. Inherited mutations in the BRCA1 gene predispose carriers to early-onset tumourigenesis and an up to 87% cumulative lifetime risk of developing breast cancer and/or ovarian cancer[4]. BRCA1-defective breast cancers are usually high grade and have poor prognoses. Moreover, ~48–66% of BRCA1 mutation carriers develop triple-negative breast cancer (TNBC), a rate that is much higher than that of non-carriers (~20%)[5–7].

BRCA1 is crucial for multiple biological processes, including DNA damage repair, cell-cycle checkpoints, ubiquitination and transcriptional regulation[8]. Studies have demonstrated that loss of BRCA1 results in defective DNA damage repair, abnormal centrosome duplication, G2-M cell-cycle checkpoint defects, growth retardation, increased apoptosis, genetic instability and tumourigenesis[9–11]. Using mouse models, we and others have demonstrated that complete knockout of Brca1 in the whole body ($Brca1^{-/-}$) causes lethality at embryonic day 7–8 (E7-8)[12,13]; in contrast, mammary-specific deletion of exon 11 of Brca1 ($Brca1^{Co/Co}$; MMTV-Cre) results in mammary tumour formation accompanied by massive genomic alterations and cellular lethality[14]. These findings prompted us to hypothesise that tumourigenesis triggered by Brca1 deficiency must encounter a lethal block that retards tumour progression, at least in early stages. Therefore, Brca1 deficiency is a double-edged sword, i.e., genome instability dysregulates massive tumour suppressor and oncogenic factors to promote tumourigenesis, but too much DNA damage initiates a lethal block by inducing apoptosis to retard tumour formation.

A genetic approach by directly breeding mutant mice carrying various $Brca1^{\Delta11/\Delta11}$ mutations has indicated that loss of function of Trp53, Atm, Chk1 or Chk2 suppresses the embryonic lethality caused by Brca1 deficiency and accelerats tumourigenesis to varying degrees[15–17]. Although many oncogenic drivers for tumourigenesis remain elusive, because the absence of Brca1 initially triggers the lethal block by inducing mitotic catastrophe and apoptosis, researchers have hypothesised the existence of secondary "hits" to modifying some cancer drivers (oncogenes or tumour suppressors) to attenuate this block, allowing Brca1-mutant cells to overcome mitotic catastrophe and ultimately resulting in mammary tumourigenesis[18]. Identifying these second "hits" may benefit clinical treatment by rebooting mitotic catastrophe after modifying the target gene or pathway.

The Sleeping Beauty (SB) DNA transposon system has been used to identify genes involved in multiple types of cancers[19–21]. This system consists of a conditionally expressed transposase and mutagenic transposon allele flanked by inverted/direct repeats. The SB transposon can be adapted to initiate mutagenesis through random insertions to cause loss or gain of gene function while tagging potential cancer driver genes. SB transposition can also be controlled to induce mutations in a specific target tissue by restricting conditional expression of the SB transposase via tissue-specific expression of Cre to avoid potential side effects. Together with high-throughput sequencing, these methods can rapidly identify cancer driver genes and related pathways, providing insight into human cancers through the use of mouse models.

In our efforts to identify these cancer drivers, we have conducted a functional-based driver gene screen by introducing the SB system ($SB+;T2Onc3+$)[22] into the Brca1 mammary-specific knockout mouse model and identified 169 candidate genes correlating with tumourigenesis. Further analysis identifies Notch1 as a top putative oncogene that overcomes apoptosis caused by

Brca1 deficiency and promotes TNBC formation by activating the epithelial–mesenchymal transition (EMT) signalling pathway.

## Results

**SB mutagenesis enhances Brca1-mutant mammary tumourigenesis.** We introduced the SB system into $Brca1^{Co/Co}$;WAP-Cre mice or $Brca1^{Co/Co}$;MMTV-Cre mice to generate four experimental cohorts of mice by interbreeding them with two independent SB transposase-T2Onc3 transposon mouse lines: SB; T2Onc3-12740 and SB;T2Onc3-12775[22] (Fig. 1a; Supplementary Fig. 1a). The mouse strains were $Brca1^{Co/Co}$;WAP-Cre;SB; T2Onc3-12740 (BrWSB40, $n = 188$), $Brca1^{Co/Co}$;WAP-Cre;SB; T2Onc3-12775 (BrWSB75, $n = 129$), $Brca1^{Co/Co}$;MMTV-Cre;SB; T2Onc3-12740 (BrMSB40, $n = 34$), and $Brca1^{Co/Co}$;MMTV-Cre; SB;T2Onc3-12775 (BrMSB75, n = 36). $Brca1^{Co/Co}$;Wap-Cre (BrW, $n = 62$) mice and $Brca1^{Co/Co}$;MMTV-Cre (BrM, $n = 56$) mice without the SB transposase or transposon were used as controls (Supplementary Data 1). The mice were monitored twice a week for tumourigenesis, and tumours were collected when they reached ~1–2 cm in diameter or the mice were moribund. Complete necropsy was performed to assess primary and metastatic tumours.

A total of 22/62 (35.5%) BrW mice developed mammary tumours in the study period of approximately 100 weeks. The SB tumourigenesis system markedly accelerated tumourigenesis and increased tumour incidence to almost 100% and tumour burden/mouse in both the 12740 and 12775 strains (Fig. 1b, c). Similarly, 26/56 (46.4%) BrM mice and all ($n = 70$) BrMSB mice developed mammary tumours during the same period of time, with a faster acceleration of tumourigenesis in BrMSB75 than in BrMSB40 strains (Supplementary Fig. 1b, c; Supplementary Data 1). The data showing that mammary tumourigenesis was accelerated in both $Brca1^{Co/Co}$;WAP-Cre and $Brca1^{Co/Co}$;MMTV-Cre mice by two independent mammary-specific SB transposition strains underscore the idea that secondary "hits" are required to attenuate the lethal block in Brca1-deficient mammary epithelial cells.

Next, we conducted molecular subtyping for tumours from these mice using Hematoxylin and eosin (H&E) and immunohistochemistry (IHC) staining for TNBC markers (Fig. 1d). Tumours from both the BrWSB and BrW groups demonstrated comparable diverse histology[18] (Supplementary Fig. 1d). The tumours of both groups also had a similar incidence of TNBC, i.e., 57% (85/149) and 50% (5/10), respectively (Fig. 1e), which is also comparable to the incidence of TNBC in human BRCA1-related breast cancers[5]. Furthermore we analysed 24 tumours from BrM mice, and 45.8% (11/24) were TNBC (Supplementary Fig. 1e). These data indicate that SB-mediated insertional mutagenesis in Brca1 mammary gland-specific knockout mice accelerates mammary tumour formation without changing the total tumour spectrum.

**Identification of driver genes in Brca1-deficient tumours.** To identify genes involved in promoting mammary tumourigenesis, we performed high-throughput sequencing for transposon insertion sites of 306 SB tumours from 245 mice (Supplementary Data 1). Common insertion sites (CISs) were identified using the TAPDANCE method, as previously described[23], which led to the identification of over 17,626 non-redundant transposon insertion sites among the SB strains 12740 and 12775 (Supplementary Data 2). Analysis of CIS distribution patterns revealed no obvious bias between the different strains (i.e., 12740 or 12775 with MMTV-Cre or WAP-Cre) for all chromosomes, including chromosomes 9 and 12, which are the original sites of the transposons in strains 12740 and 12775, respectively

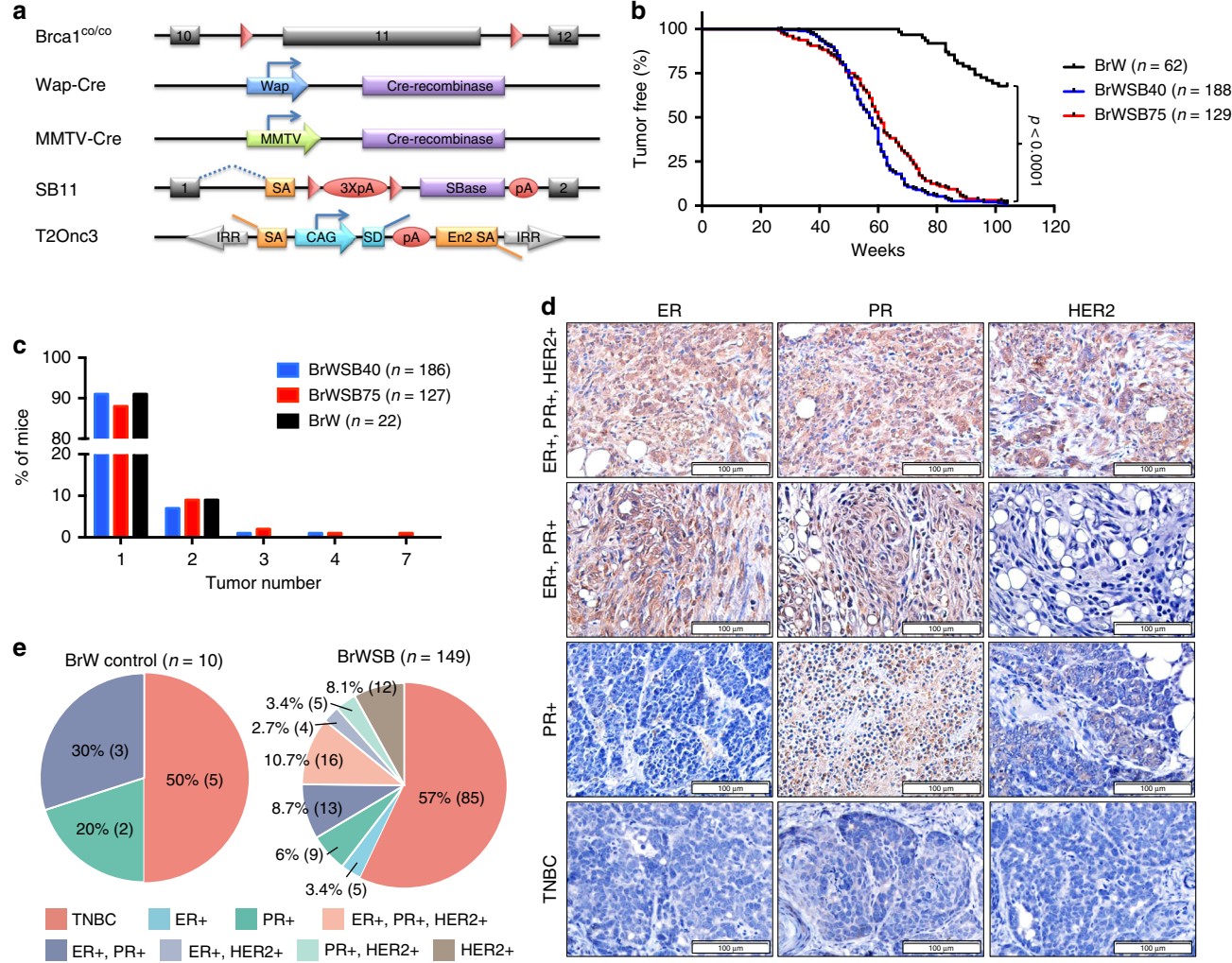

**Fig. 1 SB mutagenesis promotes tumourigenesis in Brca1 mammary-specific knockout mice. a** Overview of the engineered alleles in mutant mice. **b** Kaplan–Meier curve showing the mammary tumour-free rate for the indicated genotypes. BrWSB40 ($n = 188$) and BrWSB75 mice ($n = 129$) showed increased tumourigenesis compared to BrW control mice ($n = 62$). BrWSB40 versus BrW ($p < 0.0001$); BrWSB75 versus BrW ($p < 0.0001$); BrWSB40 versus BrWSB75 ($p = 0.9984$) by log-rank tests with GraphPad Prism. **c** Numbers of tumours per mouse in different groups. **d** IHC staining with antibodies against ER, PR and Her2. **e** TNBC incidence among different groups.

(Supplementary Fig. 2a–c). Therefore, all insertion sites were further analysed.

Using a cut-off point of their appearance in at least 5% of tumours in the 12740 and 12775 strains, we identified 119 putative driver genes (Supplementary Data 3) from the BrWSB group and 90 genes from the BrMSB group (Supplementary Data 4). Combining the two groups together yielded 169 distinct genes with 40 overlapping genes (Fig. 2a), which appeared in 6–36% of all tumour samples (Fig. 2b).

We hypothesised that other genes, in addition to the 40 overlapping genes, might be involved in accelerating tumourigenesis. Therefore, we conducted functional enrichment analysis using the KEGG and GO database, as well as gene set enrichment analysis (GSEA), for all 169 candidate genes. Some functional items were enriched, including ubiquitin-mediated proteolysis and cell adhesion-related functions, (Supplementary Data 5–7). We also conducted protein–protein interaction analysis for all 169 candidate genes. The results highlighted several clusters, including the ubiquitination system, cytoskeleton and cell junction, gene expression regulation, and protein kinase/phosphatase modification-related functions (Fig. 2c).

**Activated Notch1 is an oncogenic driver for tumourigenesis.** We divided the candidate drivers into oncogenes and tumour suppressors based on the insertion patterns of the T2Onc3 transposon (i.e., whether the insertions are clustered at hot-spot regions or widely distributed and in the same or opposite transcriptional direction of the host gene) (Fig. 3a, Supplementary Data 8). Among all 169 potential cancer drivers, 33 genes, including *Notch1, Jup* and *Met*, showed oncogenic features (Supplementary Fig. 3a), whereas the majority of genes, including *Lipc, Cntn5, Hdac4, Nrxn3* and *Cntnap5c*, exhibited tumour suppressor features (Supplementary Fig. 3b).

Notch1 is a large transmembrane protein that is associated with several human malignancies. Interaction of Notch1 with its ligands will trigger two sequential proteolytic cleavages of Notch1 to release ICN1 (intracellular domain of Notch1), allowing it to be translocated to the nucleus to participate in transcriptional regulation[24]. Although the role of Notch1 in cancer formation has been well established, it is somewhat surprising that it can act as an oncogene or a tumour suppressor in different contexts[25,26]. Thus, we decided to focus on Notch1 to illustrate the mechanism underlying its oncogenic function. Of 138 insertion sites identified in 108 Notch1-trapped SB tumours, 136 (98.6%) were

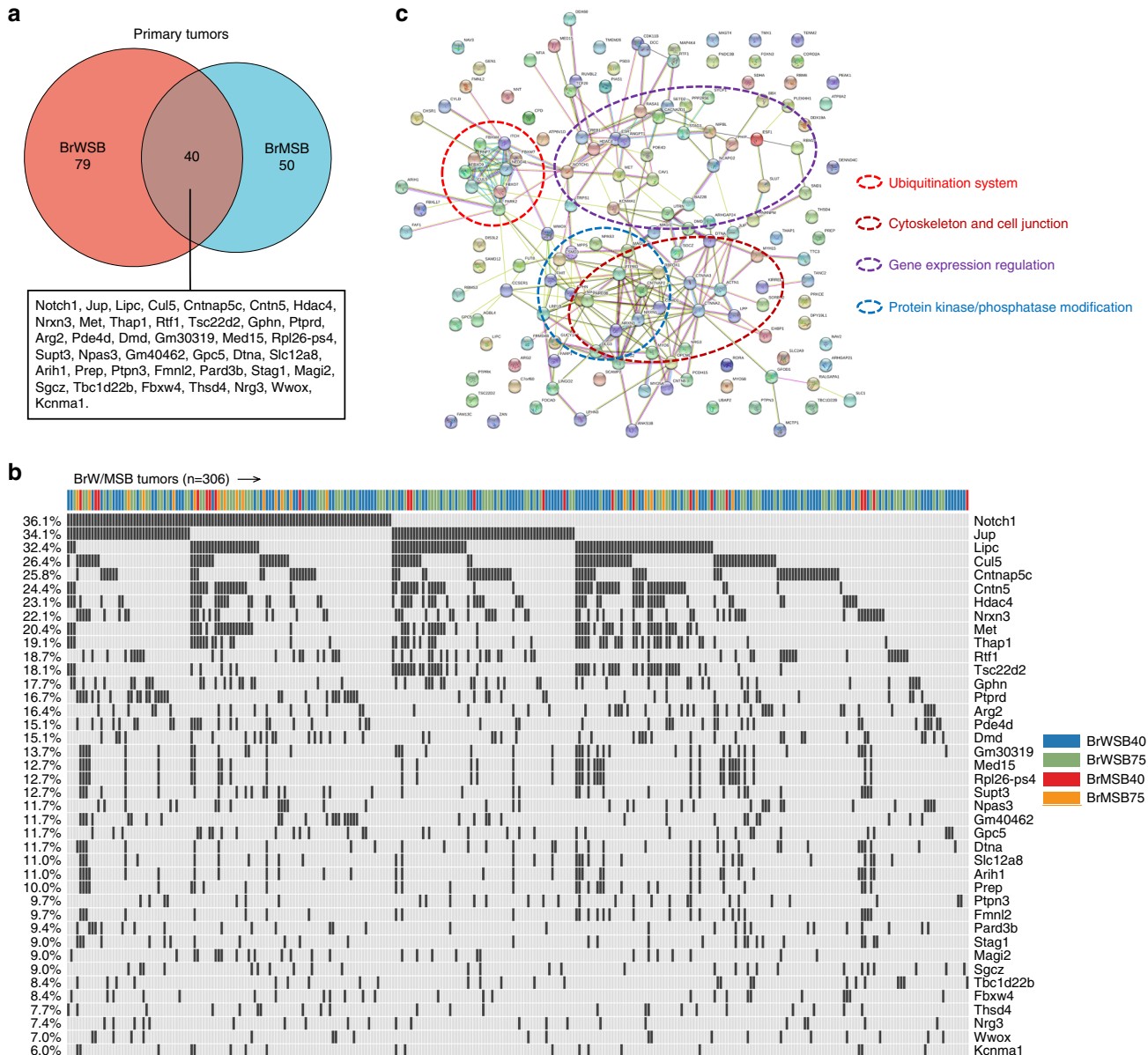

**Fig. 2 SB-driven candidate gene identification and functional annotations. a** Venn diagram indicating CIS genes for the BrWSB and BrMSB groups. There were 40 genes shared between the two groups. **b** Oncoplot of the 40 genes showing their frequency in all tumour samples. **c** Protein–protein interaction analysis of SB-driven candidate genes using STRING online tools.

concentrated between exon 25 and exon 30 (Fig. 3b), and 73.5% (100/136) were in the same direction as the direction of gene expression, indicating that the transposon specifically inserted to result in overexpression of ICN1. RNA-sequencing data also demonstrated that overexpression of Notch1 starts from exons 25–27 (Fig. 3c), which is consistent with activated Notch1 protein expression analysis (Fig. 3d). This finding suggests an oncogenic role for Notch1 activation in Brca1-mutant tumourigenesis (Supplementary Fig. 4a).

In addition, we designed two shRNAs targeting endogenous Notch1 (the N terminus of Notch1, shRNA-N) or pan-Notch1, including both SB transposon-driven Notch1 and endogenous Notch1 (the C terminus of Notch1, shRNA-ICN) (Supplementary Fig. 4a, b), and applied them to cell lines derived from Notch1-driven SB tumours (MK1370-3R) and cell lines derived from non-Notch1-driven SB tumours (MK1097-5R) (Fig. 3e, Supplementary Fig. 4c, d). Knockdown of ICN1 blocked the growth of MK1370-3R tumours, whereas targeting endogenous

Notch1 did not have a significant effect (Fig. 3e, f). In contrast, targeting either the N terminus or C terminus of Notch1 in MK1097-5R tumours did not result in any obvious effects (Fig. 3e, g). These data confirm that transposon insertion-induced overexpression of ICN1 is the driver for MK1370-3R tumours but not for MK1097-5R tumour, which may be driven by other factors. Consistently, we found that tumours with Notch1 activation exhibited faster progression than did non-Notch1-driven tumours (Fig. 3h).

We previously showed that Brca1 deficiency results in cell death that can be repressed by p53 loss[15]. We suspected that Notch1 activation might also affect cell lethality to attenuate the lethal block. To examine this possibility, we generated a Brca1 conditional knockout ES cell line ($Brca1^{Co/-}$;Cre-ERT2), in which Cre-mediated recombination is controlled by 4-hydratamoxifen (4-HT) from our $Brca1^{fl/Exon11}$ ES cells[27]. These cells enabled us to compare the impact of Notch1 activation or p53 loss on the viability of Brca1-deficient cells (Supplementary Fig. 4e, f). Acute knockout of Brca1

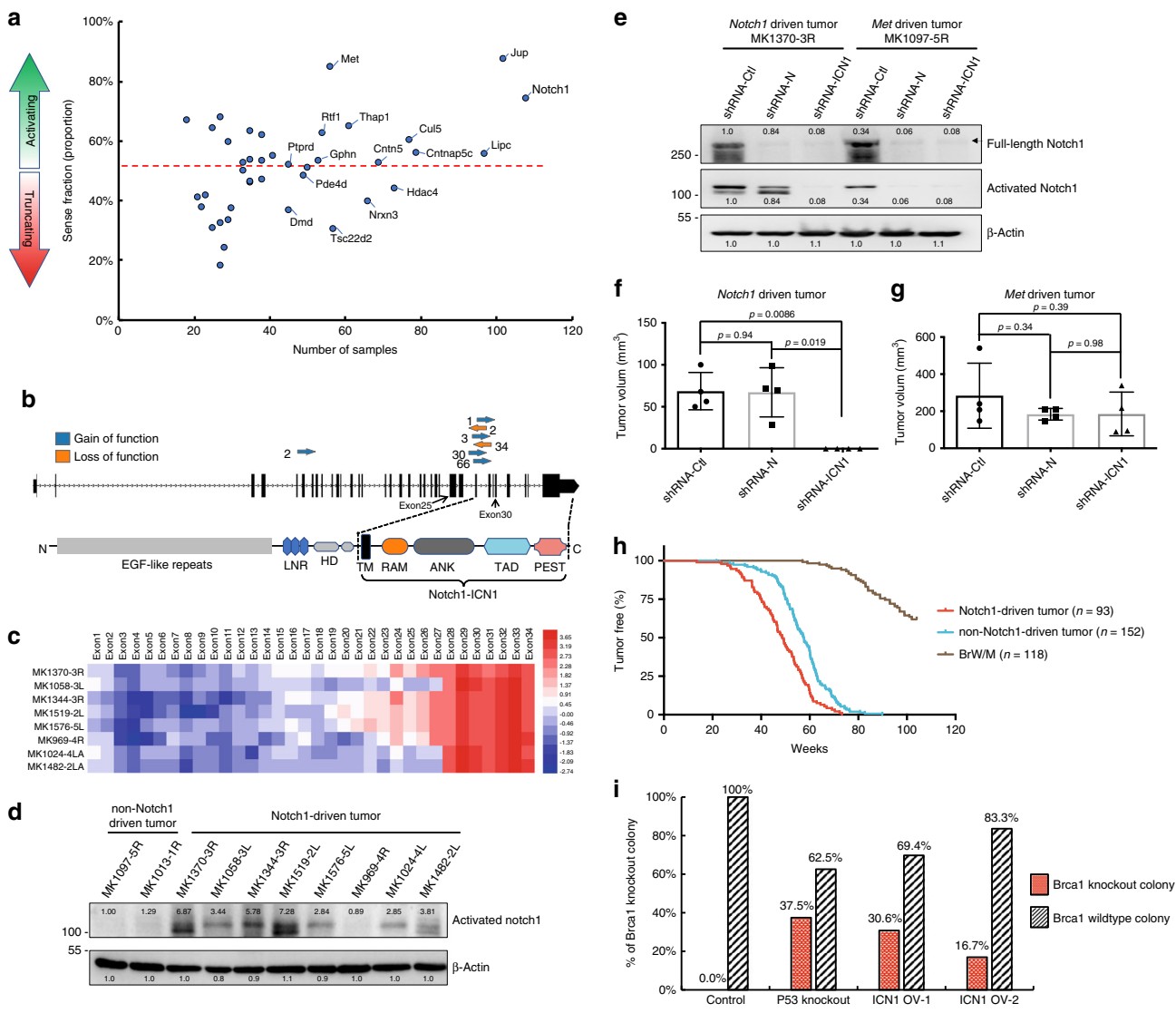

**Fig. 3 Effect of Notch1 activation on cell death and tumourigenesis. a** Predicted effect of candidate genes, as indicated by their sense fraction of insertions based on the direction of the CAG promoter and the transcriptional direction of the inserted gene. Genes with a strong bias towards sense insertions are expected to be activated and might serve as oncogenic drivers. Conversely, those biased towards antisense insertions are predicted to be inactivated or yield truncated products and serve as tumour suppressors. The dashed line in the middle indicates the equal ratio of sense and antisense insertions. **b** Structure of Notch1 and transposon insertion sites within the *Notch1* gene. Blue arrows indicate that the promoter in the transposon is in the same orientation as the host gene, and red arrows indicate different directions. Insertion frequencies are indicated by numbers. **c** Expression analysis of all exons of the *Notch1* gene based on RNA-seq of Notch1-driven SB tumours. **d** Activated Notch1 protein expression analysis based on western blotting compared with non-Notch1-driven tumours. **e** Western blot analysis of Notch1-driven tumours (MK1370-3R) and non-Notch1-driven tumours (MK1097-5R) after shRNA knockdown. **f, g** Tumour volume measured at day 30 of MK1370-3RMT (**f**) and MK1097-5RMT (**g**) after shRNA-mediated knockdown of endogenous Notch1 or pan-Notch1; $n = 4$ biologically independent animals. The $t$-test was used to determine the significance of the difference between the different sets of data. **h** Kaplan–Meier curve showing the mammary tumours-free rate for SB mice with Notch1-driven tumours ($n = 93$) and non-Notch1-driven tumours ($n = 152$), as well as BrW ($n = 62$) and BrM ($n = 56$) control mice. Notch1-driven tumours tended to show earlier onset compared with non-Notch1-driven tumours ($p < 0.0001$) by the log-rank test. **i** Percentage of different types of ES colonies with wild-type Brca1 and knockout after treatment with 4-HT to induce Brca1 knockout in p53 mutant, ICN1-overexpressing and parental ES cell lines. Cells were inoculated with 4-HT for 3 days (to delete Brca1), followed by disassociation and replating on feeder cells with regular medium. Single ES colonies were selected 7 days later for genotyping of their Brca1 status. Data are presented as mean values ± s.d.

by 4-HT treatment resulted in no *Brca1*$^{-/-}$ ES colonies ($n = 36$); we observed 37.5% (18/48) of *Brca1*$^{-/-}$ ES colonies among p53-knockout ES cells (*Brca1*$^{Co/-}$;*Cre-ERT2*;*p53*$^{-/-}$). In two independently derived ES cell lines overexpressing ICN1 (*Brca1*$^{Co/-}$;*Cre-ERT2*;*ICN1*$^{OV}$), acute knockout of Brca1 after administration of 4-HT yielded 16.7% (6/36) and 30.6% (11/36) *Brca1*$^{-/-}$ ES colonies (Fig. 3i), indicating that activation of Notch1 suppressed, at least in part, the lethality caused by Brca1 deficiency.

Overall, these data suggest that ICN serves as an oncogene to promote the initiation of Brca1-associated mammary tumourigenesis, which may occur by suppressing the lethality effect caused by Brca1 deficiency.

**Notch1 suppresses BRCA1 defects caused mitotic catastrophe.** To explore how ICN1 suppresses the lethality effect under Brca1-deficient conditions, we used a tet-on system to induce ICN1

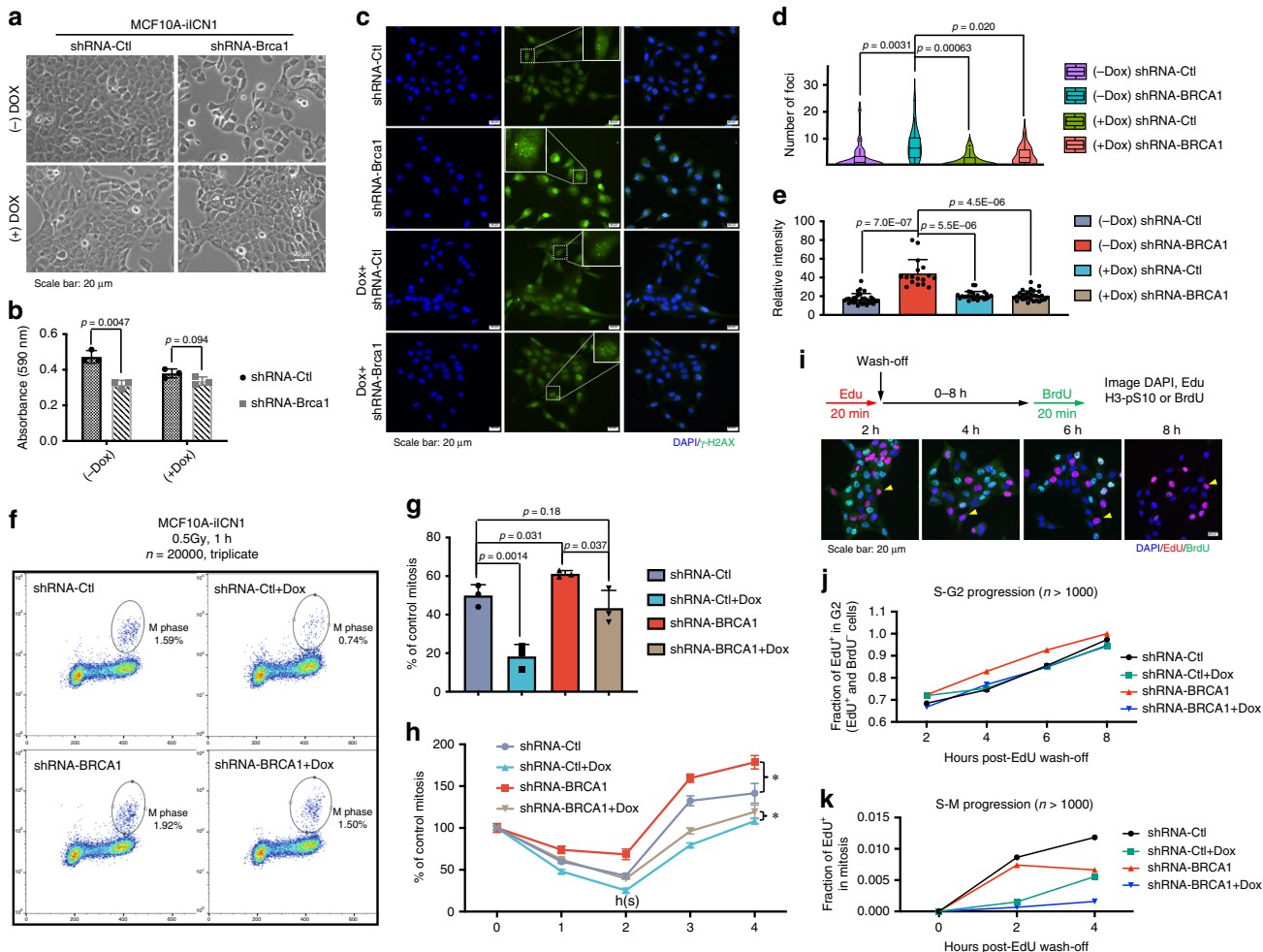

**Fig. 4 Notch1 activation suppresses the lethality caused by BRCA1 deficiency via cell-cycle regulation. a** Overexpression of ICN1 suppressed the cell death caused by BRCA1 acute knockdown in MCF10A naïve cells. **b** Quantification analysis by MTT assays regarding the rescue effect of ICN1 on BRCA1 deficiency. Analysis on the 3rd day after Dox and/or lentivirus with shRNA-BRCA1 induction; $n = 3$ biologically independent experiments. **c** IF staining and foci counting **d** and intensity quantification **e** of γ-H2AX to indicate DNA damage at 48 h after BRCA1 acute knockdown with or without ICN1 overexpression; $n = 17–30$ independent cell measurements. **f** Mitotic index analysis of MCF10A cells at 1 h after 0.5 Gy gamma-irradiation treatment. A total of 20,000 cells were counted, and three repeats were used to determine the SD. **g** Quantification analysis of the relative M-phase portion in the different treatment groups in **e**; values were normalised to the no-irradiation treatment value for the individual groups; $n = 3$ biologically independent experiments. The p-value was determined by the t-test with one-tailed distribution. **h** Dynamic changes in the mitotic index at different time points after 0.5 Gy gamma-irradiation treatment; $n = 3$ biologically independent experiments. * indicates $p < 0.05$. **i** Pulse-chase assay to measure S-G2 and S-M progression. The yellow arrowhead indicates representative cells that progressed to G2 phase at a particular time point. **j** Fraction of EdU-positive cells entering G2 phase (BrdU-negative) during the chase ($n = 1800$). **k** S-M progression indicated by EdU and p-H3 double-positive cells ($n = 1100$). Data are presented as mean values ± s.d.

overexpression in MCF10A cells by doxycycline (Dox) administration (Supplementary Fig. 5a–c). Acute knockdown of BRCA1 by using shRNA led to cell death, whereas Dox-induced ICN1 overexpression suppressed this lethal effect (Fig. 4a, b). Our further analysis detected markedly increased γ-H2AX and 53BP1 levels upon acute knockdown of BRCA1, but overexpression of ICN1 blocked this effect (Fig. 4c–e, Supplementary Fig. 5d, e), suggesting that activation of NOTCH1 abolishes the mitotic catastrophe caused by BRCA1 deficiency by decreasing DNA damage. The rescue effect was also observed when we treated the cells with the NOTCH1 ligand Jagged1 (Supplementary Fig. 5f, g). Moreover, the cell growth rate was decreased after ICN1 overexpression before cell confluence at day 5 (Supplementary Fig. 5h); and the stall in growth was maintained after replating the cells (Supplementary Fig. 5i). By synchronising the cell cycle with thymidine, we found that overexpression of ICN1 caused a

G2/M-phase lag (Supplementary Fig. 5j, k), even with BRCA1 knockdown.

We previously showed that BRCA1 deficiency impairs the G2/M cell-cycle checkpoint, which causes severe genomic instability[28]. Thus, we investigated whether ICN1 is able to restore the G2/M checkpoint, which accounts for the decreased DNA damage. By assessing the mitotic index (MI) with ICN1 overexpression and/or BRCA1 knockdown after gamma-irradiation (IR), we found that ICN1 decreased the mitotic population in both parental and BRCA1-knockdown cells after DNA damage in both MCF10A (Fig. 4f, g), and T47D cells (Supplementary Fig. 5l, m). These data suggest a role for ICN in activation of the G2/M cell-cycle checkpoint upon irradiation, which is sufficient to override the lethal block caused by BRCA1 deficiency. We then conducted a time course experiment after IR, and the data also indicated that ICN significantly reduced the

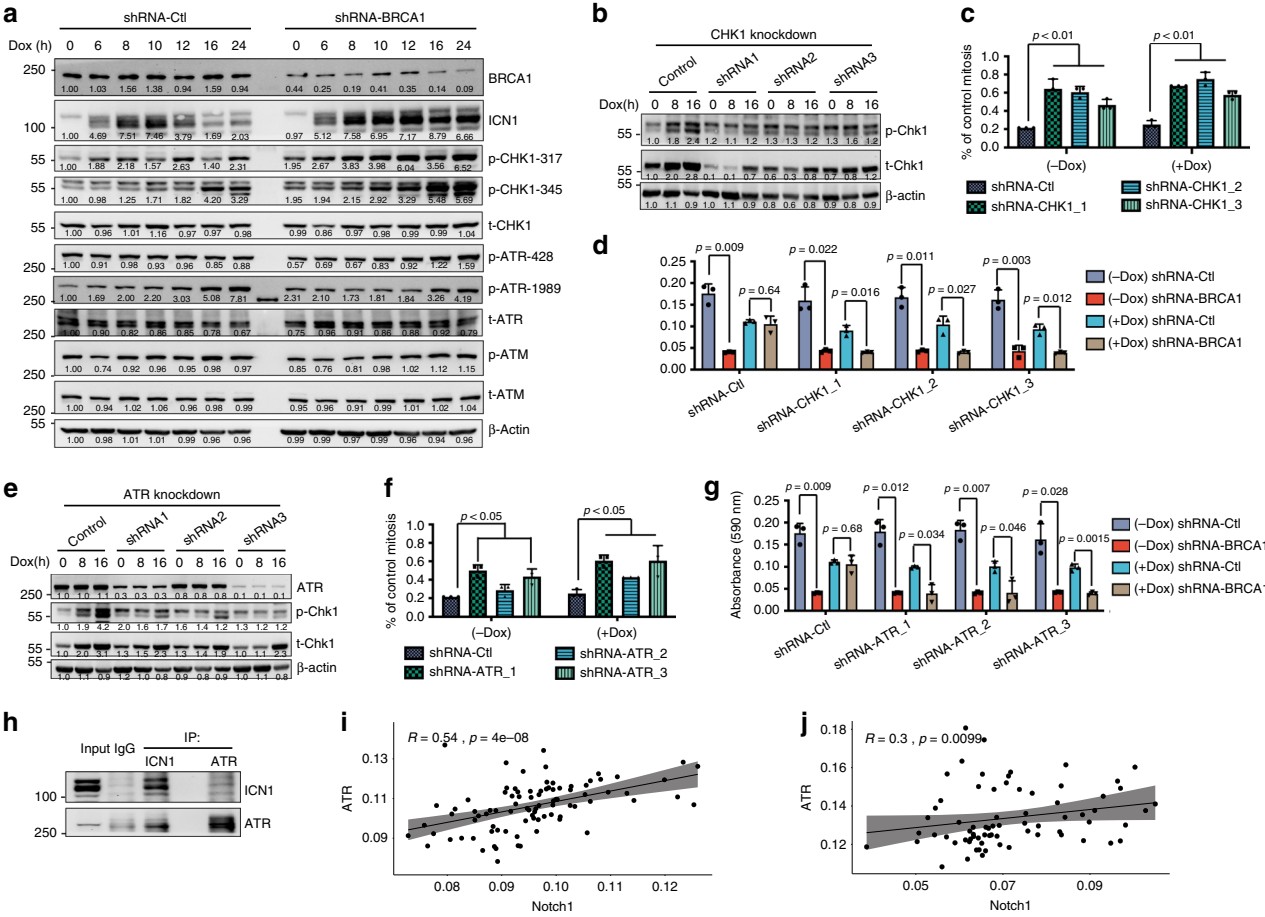

**Fig. 5 Notch1 activation rescues the lethal effect caused by BRCA1 deficiency through the ATR–CHK1 axis. a** Western blotting analysis of cell-cycle checkpoint proteins at different time points after Dox administration. **b** Western blotting analysis of CHK1 phosphorylation after ICN1 induction after knockdown of CHK1. **c** MI assay of MCF10A cells when CHK1 was knocked down with/without ICN1 overexpression. Cells were collected at 1 h after 0.5 Gy irradiation treatment; $n = 3$ biologically independent experiments. **d** MTT assay evaluating the rescue effect of Brca1 acute knockdown on the 3rd day when CHK1 was knocked down with shRNA; $n = 3$ biologically independent experiments. **e** Western blotting analysis of CHK1 phosphorylation after ICN1 induction following ATR knockdown. **f** MI assay of MCF10A cells when ATR was knocked down with/without ICN1 overexpression. Cells were collected at 1 h after 0.5 Gy irradiation treatment; $n = 3$ biologically independent experiments. **g** MTT assay evaluating the rescue effect of Brca1 acute knockdown on the 3rd day after ATR knockdown with shRNA; $n = 3$ biologically independent experiments. **h** IP analysis indicated that ICN1 can directly bind with ATR. **i**, **j** Immunohistochemistry staining of a human TNBC patient tissue microarray for target proteins. The scatter plots indicate a positive correlation between ICN1 and p-ATR ($n = 90$ for **i**, $n = 72$ for **j**); $R$ and $p$-values were obtained based on Spearman's rank correlation coefficient $R$'s and probability ($p$) value calculator. Data are presented as mean values ± s.d.

population of MI in BRCA1 WT and knockdown cells at multiple time points (Fig. 4h, Supplementary Fig. 5n).

Next, we specifically evaluated S-G2 and S-M progression using quantitative image-based cytometry combined with a pulse-chase assay (Fig. 4i). The fraction of EdU-positive cells that progressed to G2 phase (BrdU-negative) during the chase indicated that BRCA1 deficiency resulted in premature entry into mitosis and that hyperactivation of NOTCH1 restored cell-cycle regulation (Fig. 4j). Meanwhile, ICN1 also markedly delayed S-M progression (Fig. 4k).

To provide further evidence, we performed western blotting for key proteins involved in cell-cycle checkpoints, and the results showed increased CHK1 phosphorylation at Ser317 and Ser345 along with ICN1 induction in both parental and BRCA1-knockdown cells to varying degrees (Fig. 5a). To investigate the function of p-CHK1, we knocked down CHK1 with shRNA (Fig. 5b) and found that it increased MI (Fig. 5c) and abolished the suppressive effect of ICN1 on the lethality caused by acute knockdown of BRCA1 (Fig. 5d). These data suggest that activated NOTCH1 functions through p-CHK1 to

restore the S/G2 or G2/M checkpoint to suppress apoptosis caused by BRCA1 defects.

The CHK1 kinase acts downstream of the ATR/ATM kinase, which is essential for embryonic development and tumour suppression[29,30]. To determine whether CHK1 phosphorylation is mediated by ATR or ATM after ICN1 overexpression, we examined the expression levels of these two kinases and found that ATR phosphorylation was moderately increased but that ATM phosphorylation was not (Fig. 5a). In addition, ATR knockdown blocked ICN1-triggered CHK1 phosphorylation (Fig. 5e), increased MI (Fig. 5f), and, similar to CHK1 knockdown, abolished the ability of ICN1 to suppress the apoptosis caused by acute knockdown of BRCA1 (Fig. 5g). However, such effects were not found when ATM was inhibited (Supplementary Fig. 6a). These observations indicate that ICN1 regulates CHK1 phosphorylation through ATR but not ATM. Immunofluorescent staining results showed that ICN1 induction markedly increased p-ATR as well as p-CHK1 localisation to the nucleus (Supplementary Fig. 6d, e). Furthermore, immunoprecipitation analysis demonstrated physical binding between ICN1 and ATR (Fig. 5h),

indicating that NOTCH1 activation modulates p-ATR localisation to regulate the ATR–CHK1 pathway; these findings may need further validation. Moreover, ATR–CHK1 activation by ICN1 was also detected in T47D cells, as well as in MCF10A cells treated with the NOTCH1 ligand Jagged1 (Supplementary Fig. 6b, c), suggesting a general feature of NOTCH1 to regulate the cell cycle in breast cancer cells.

To further confirm the correlation between ICN1 and p-ATR, we used two sets of human TNBC patient tissue microarrays to conduct immunohistochemistry staining (Supplementary Fig. 6f). The results showed that the p-ATR protein level correlated significantly with the NOTCH1 activation level (Fig. 5i, j); conversely, other proteins did not show a consistent correlation with NOTCH1 (Supplementary Fig. 6g, h).

Taken together, these data reveal that BRCA1 deficiency impairs the S/G2 and G2/M cell-cycle checkpoints, which triggers the lethal block, but that ICN1 overexpression restores these checkpoints and suppresses the mitotic catastrophe caused by BRCA1 deficiency through a non-canonical target: the ATR–CHK1 axis.

**NOTCH1 correlates with increased human TNBC formation.** BRCA1 mutation increases the formation of malignant TNBC in humans[5]. To identify the driver genes responsible for *BRCA1*-deficient TNBC formation, we used comparative coefficient analysis to select candidates correlating closely with TNBC incidence. First, we converted 169 candidate genes to human genes based on mouse-human orthologues. Then, we traced the expression data and corresponding pathology information for each gene from the human clinical databases TCGA[31] and METABRIC[32] and separated the patients into defined equal cohorts based on the expression level of candidate genes. Finally, the TNBC percentage was calculated for each cohort to determine the correlation between gene expression and TNBC incidence (Fig. 6a, Supplementary Fig. 7).

Fifteen genes in our 169-gene pool showed good correlations with TNBC patients in both TCGA and METABRIC databases (Fig. 6b). *NOTCH1, ARHGAP21, FMNL2* and *TCF20* correlated positively with TNBC incidence, i.e., patients with high expression of these genes tended to develop TNBC. Expression pattern analysis of the *NOTCH1* gene indicated that TNBC and basal-type tumours exhibited significantly increased levels of NOTCH1, regardless of whether BRCA1 was mutated (Fig. 6c). TNBC and basal tumour incidence markedly increased with increasing NOTCH1 expression (Fig. 6d). In 54 BRCA1-deficient breast cancers in this cohort, increased expression of NOTCH1 also correlated with TNBC and the basal-type. Analysis of protein levels in BRCA1-deficient tumours indicated that a high protein expression level of NOTCH1 correlated with TNBC incidence (Supplementary Fig. 8a, b). Moreover, under BRCA1-defective conditions, NOTCH1 tended to display an enhanced ability to promote TNBC and basal-type breast cancer (Supplementary Fig. 8c, d). A Kaplan–Meier survival curve indicated high expression levels of NOTCH1 to be associated with worse outcomes in TNBC patients (Supplementary Fig. 8e, f).

To investigate whether Notch1 activation in our SB tumours also contributes to TNBC formation, we conducted IHC staining for ER, PR and Her2, The data demonstrated that 74.24% (49/66) of ICN1-driven SB tumours were TNBC (Supplementary Fig. 9a), which was much higher than the average percentage of BrWSB (57%) and BrW control (50%) tumours (Fig. 1e). Moreover, we observed that 62.5% (20/32) of ICN1-driven SB tumours were the basal-type, compared with 40% (8/20) of randomly selected non-ICN1-driven SB tumours (Supplementary Fig. 9b), suggesting that ICN1 also enhances basal-type tumour formation.

In conclusion, our results indicate that NOTCH1 may be an oncogenic driver for BRCA1-related TNBC and basal-type tumours.

**NOTCH1 activation stimulates EMT to promote TNBC formation.** To further explore the related mechanism, we conducted gene expression analysis of Notch1-driven TNBC tumours compared with non-Notch1-driven TNBC tumours (Fig. 7a). Gene set enrichment analysis (GSEA) demonstrated that genes upregulated in the EMT process were highly expressed in Notch1-driven tumours (Fig. 7b, Supplementary Data 9). In contrast, genes downregulated during EMT revealed low levels of expression in Notch1-driven tumours (Fig. 7c, Supplementary Data 9). These results indicate that EMT-related genes correlated strictly with Notch1 activation in TNBC tumours. It has been reported that Notch1 signalling regulates the EMT in human breast cancer[33,34], and that expression of EMT-related markers in TNBCs can serve as a signature of a certain subgroup of TNBC[35,36]. Therefore, we studied whether NOTCH1 promotes EMT transition, leading to TNBC formation in BRCA1-defective conditions.

By using a tet-on system to induce ICN1 overexpression in different mammary cell lines, we found that activated NOTCH1 upregulated Fibronectin 1, Vimentin and Slug to varying degrees in MCF10A, T47D and MCF7 cells (Fig. 7d, e, Supplementary Fig. 9c, d). Moreover, we induced Notch1 expression in ATR or CHK1 knockdown cells and then assessed their EMT status. As shown in the Supplementary Fig. 9e, f, Dox-mediated Notch1 induction increased expression of Fibronectin; in contrast, knockdown of ATR or CHK1 did not affect expression. These data suggest that the function of Notch1 in EMT stimulation is independent of ATR or CHK1. Human BRCA1-mutant breast cancer samples also showed high levels of mesenchymal markers (Fig. 7f). Furthermore, a meta-analysis of BRCA1-related cancer revealed that all basal-like tumours belong to the TNBC subtype, much higher than the average level (Supplementary Fig. 8g, h). These data suggest the EMT might be the driver of BRCA1-mutant cell transdifferentiation. Therefore, we conclude that NOTCH1 hyperactivation promotes EMT, which might induce the formation of TNBC and basal-like breast cancer.

**Inhibition of ATR–CHK1 enhances the tumouricidal activity.** Cisplatin preferentially kills BRCA1-deficient breast cancers compared to BRCA1-proficient cancers; however, the inhibition efficiency is always compromised when cancers gain various additional alterations[37,38]. Furthermore, a high dosage of cisplatin can cause serious nephrotoxicity[39,40]. Our previous data indicate that cisplatin inhibits tumour metastasis by blocking EMT, though prolonged treatment may result in drug resistance[41].

TNBC is the most difficult type of breast cancer to treat due to its unresponsiveness to current clinical targeted therapies, high rate of recurrence and poor prognosis[42]. A high expression level of Notch1 correlates closely with TNBC incidence and activates ATR–CHK1 signalling to compensate for the DNA damage repair deficiency and benefit tumourigenesis, and cisplatin causes DNA crosslinking, leading to DNA double strand breaks (DSBs). Accordingly, we hypothesised that ATR or CHK1 inhibitor combined with cisplatin might have a synthetic lethal effect on TNBC samples. To investigate this, we first assessed the effect on the HCC1937 cell line, and the results indicated that a low dosage of an ATR inhibitor (AZD6738 or VE821) combined with different concentrations of cisplatin effectively killed HCC1937 cells (Fig. 8a, b). Notably, combined treatment with cisplatin and an ATM inhibitor (KU-60019) failed to show a synergistic effect

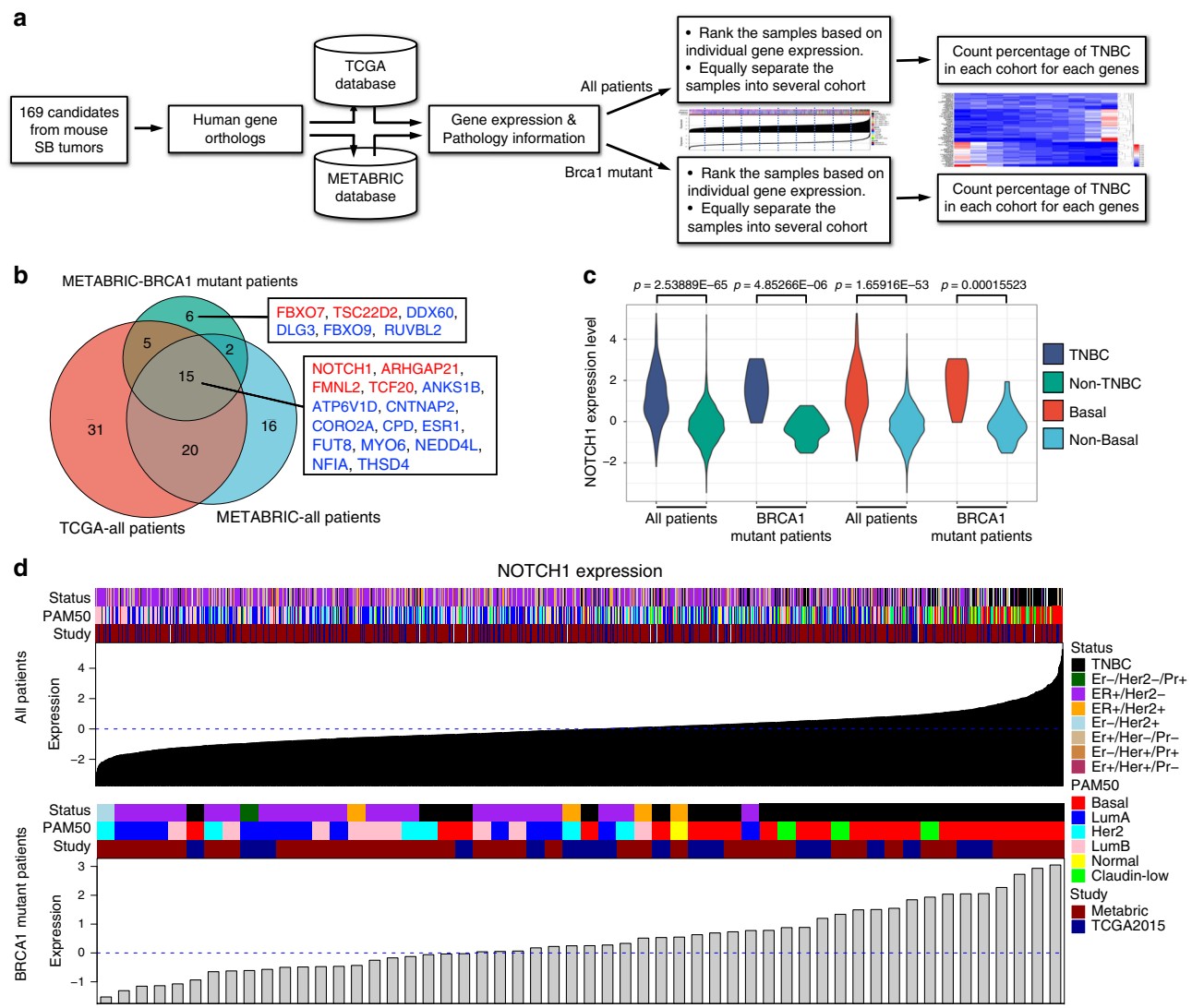

**Fig. 6 Correlation of NOTCH1 expression with human TNBC and basal-type breast cancer. a** Flowchart illustrating the strategy of TNBC correlation analysis for candidate genes. **b** Venn diagram showing TNBC-correlated genes obtained from TCGA and METABRIC databases. Genes labelled with red correlated positively with TNBC incidence, and genes highlighted with blue correlated negatively. **c** Distribution of NOTCH1 expression levels in TNBC vs. non-TNBC and basal vs. non-basal human patients. The t-test was used to determine the significance of differences between the different sets of data. **d** Correlation between NOTCH1 expression level and tumour subtype. Tumours were ranked based on NOTCH1 expression level, and tumour subtypes are labelled on the top. Data were obtained from TCGA (n = 403) and METABRIC (n = 1898) databases.

in killing cancer cells, which is consistent with our previous finding that ICN1 regulates downstream effects through ATR but not ATM. Moreover, the CHK1 inhibitor CCT245737 or LY2603618 also increased sensitivity to cisplatin (Fig. 8a, b). Next, we tested four additional BRCA1-mutant TNBC cell lines and found that the ATR or CHK1 inhibitor markedly enhanced killing but that the ATM inhibitor showed the opposite effect (Fig. 8c). These data verify our model that ICN1 can, through the non-canonical target ATR or CHK1, regulate the cell cycle and promote TNBC formation.

Furthermore, we conducted in vivo treatment of TM00091 PDX tumours, which carry two BRCA1 point mutations (Q356R and C61G), with NOTCH1 hyperactivation (Supplementary Fig. 10a). Because a high dose of cisplatin markedly decreased mouse body weight (Supplementary Fig. 10b) indicating a toxic effect, we administered cisplatin at a low dose. The results showed that AZD6738 single treatment did not affect tumour growth, even at high doses. A low dosage of AZD6738 markedly increased the killing effect of cisplatin (Fig. 8d, Supplementary Fig. 10c).

Notably, combined with ATRi, cisplatin at a low dose (1.5 mg/kg) significantly inhibited tumour progression. According to IHC staining, double treatment induced massive apoptosis/death compared with high-dosage single treatment of ATRi or cisplatin (Fig. 8e, f).

In conclusion, cisplatin-mediated inhibition of the NOTCH1 non-canonical target ATR–CHK1 cascade can synergistically kill BRCA1-related TNBC both in vitro and in vivo.

## Discussion

Tumours initiate from normal cells through a long "evolution" process, encountering various blocks, including immune checkpoints, telomere limitation, contact inhibition and genome instability-induced apoptosis, among others. Our previous studies indicated that the progression of Brca1-related tumourigenesis occurs through a long latency that is accompanied by initial cellular lethality[12,14], suggesting the existence of such blocks. In the present study, SB-mediated mutagenesis revealed several findings that advance our understanding of this situation. We

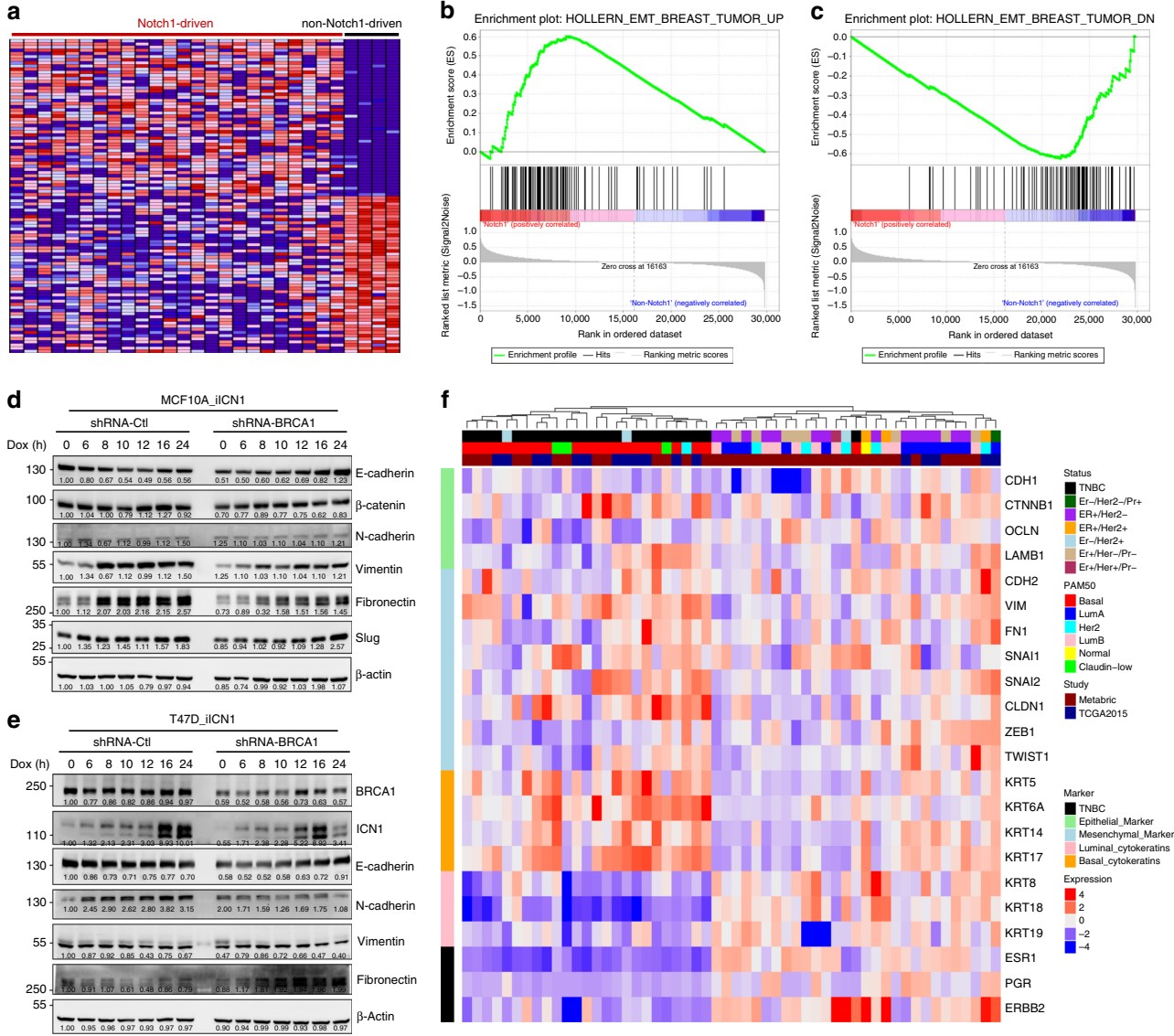

**Fig. 7 Notch1 stimulates BRCA1-related TNBC progression by promoting EMT. a** Gene expression analysis of Notch1-driven TNBC SB tumours ($n = 24$) compared with non-Notch1-driven TNBC SB tumours ($n = 4$). **b** GSEA analysis reveals that upregulated genes in Notch1-driven tumours are enriched in the EMT_breast_tumour_up term. **c** GSEA analysis indicates that downregulated genes in Notch1-driven tumours are enriched in the EMT_breast_tumour_dn term. **d** Western blotting analysis of EMT markers after induction of ICN1 in MCF10A and MCF10A-BRCA1-knockdown cells. **e** Western blotting analysis of EMT markers after induction of ICN1 in T47D and T47D-BRCA1-knockdown cells. **f** Meta-analysis of BRCA1-mutant human breast tumours for EMT markers, basal markers, luminal markers and TNBC markers.

demonstrated that a defective G2/M cell-cycle checkpoint caused by Brca1 deficiency serves as one such block, as restoration of this checkpoint by Notch1 activation suppressed lethality and allowed Brca1-mutant cells to grow. Further analysis revealed several previously unknown functions of Notch1 that may be necessary in accelerating Brca1-related TNBC tumourigenesis.

The Notch signalling pathway has an important role in several biological processes. Deregulated Notch signalling is associated with several human malignancies, such as carcinomas of the lung, colon, head and neck, breast, pancreas and kidney, either as an oncogene or a tumour suppressor[26,43,44]. In the present study, our data clearly demonstrate that expression of ICN1, representing Notch1 activation, promotes Brca1-associated mammary tumour formation. One of the major abnormalities induced by Brca1 deficiency is a defective G2/M cell-cycle checkpoint[28], which may serve as a double-edged sword that, on the one hand, causes genetic instability leading to cell death and, on the other hand,

creates an environment for mutations leading to tumourigenesis if p53 is mutated[14]. We further showed that loss of p53 enables Brca1-deficient cells to survive despite the presence of widespread genetic instability due to inactivation of apoptosis mediated by p53[15]. Notably, we found that activation of Notch1 also enables Brca1-deficient cells to survive through a distinct mechanism, namely, partial restoration of the G2/M cell-cycle checkpoint possibly through the ATR–CHK1 cascade, which is known to have an essential role in activating G2/M[29]. ATR also activates intra-S and S/G2 cell-cycle checkpoints[45,46], which prompted us to examine these checkpoints. Our data indicate that Notch1 activation does not have an obvious role in the intra-S checkpoint but that it does activate the S/G2 cell-cycle checkpoint. Activation of these checkpoints delays cell-cycle progression, as reflected by reduced cell proliferation and attenuation of the lethal block caused by BRCA1 deficiency. Clinical sample analysis demonstrated that NOTCH1 activation correlated positively with the

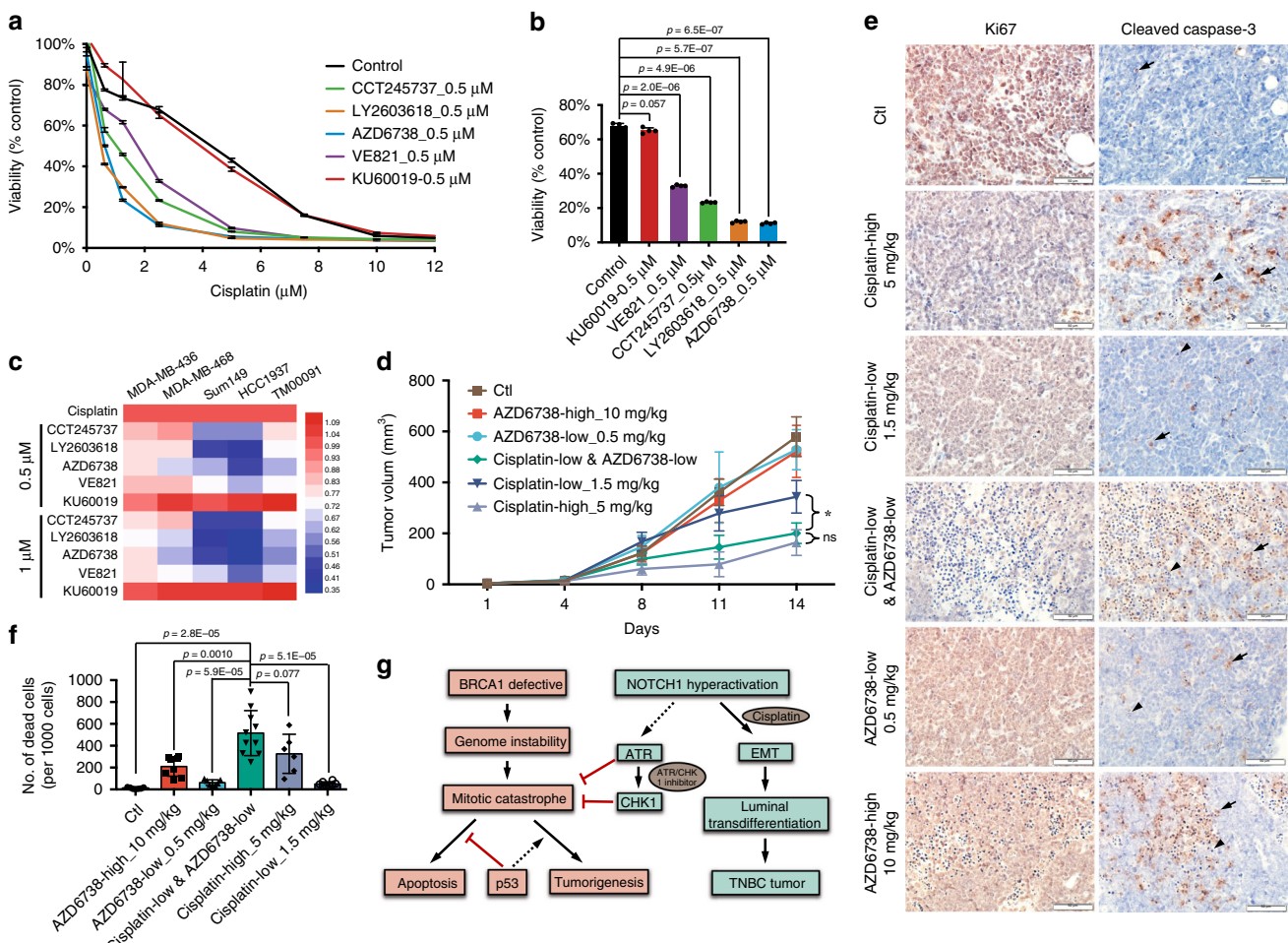

**Fig. 8 Combined drug treatment of TNBC tumours. a** Effect of combined treatment of cisplatin with ATR, ATM or CHK1 inhibitors on HCC1937 cells; $n = 4$ biologically independent experiments. **b** Quantification analysis of combined drug treatment in HCC1937 cells. Cisplatin (2.5 µM), ATR, ATM or CHK1 inhibitor (0.5 µM); $n = 3$ biologically independent experiments. **c** Killing effect on TNBC cells treated with cisplatin combined with different inhibitors. The colour in the heatmap represents the relative values of the areas under the curves. **d** In vivo effect of cisplatin combined with ATR inhibitor (AZD6738) on tumours in the PDX model (TM00091), which carries the BRCA1 mutation and highly expresses NOTCH1. Six mice were used for each treatment group. **e** IHC staining of Ki67 and cleaved caspase-3 in PDX tumours after drug treatment. Five tumours from different mice were examined for each treatment group. **f** Quantification analysis of dead cells (arrowhead), including cleaved caspase-3-positive cells (arrow) in each treatment group. Six to ten areas from different tumours were counted. **g** Illustration of the ICN1 function for TNBC formation in BRCA1-defective conditions. The t-test was used to determine the significance of differences between the different sets of data. Data are presented as mean values ± s.d. * indicates $p < 0.05$.

phosphorylation level of ATR plus the capacity of direct binding between NOTCH1 and ATR; therefore, we speculate that NOTCH1 rescues BRCA1-defective lethality through a non-canonical target, the ATR–CHK1 signalling pathway. Consequently, we conclude that NOTCH1 has an oncogenic function in the presence of BRCA1 loss. Nonetheless, we should note that under some other conditions, e.g., in tumours with intact DNA damage repair systems or cell-cycle checkpoints, activation of ATR–CHK1 signalling would inhibit cell-cycle progression and elicit tumour suppressor functions to block tumour development. This may help to explain the dual functions of NOTCH1 as a tumour suppressor and an oncogene under different circumstances[26]. Obviously, further studies are needed to support this hypothesis.

Previous studies have revealed that activation of NOTCH1 correlates with cancer stem cell maintenance and expansion[47] and that high NOTCH1 expression may serve as a prognostic marker in patients with TNBC[48], yet the underlying mechanism remains elusive. Our study demonstrates that activation of Notch1 promotes TNBC formation. BRCA1-deficient cancers

originate from luminal cells, which are largely ERα positive initially[49]. Our previous study analysing tumours at different stages also showed that Brca1[−/−] mammary tumours are initially ERα positive and gradually become negative as they grow[50]; nonetheless, it remains unclear how ERα is lost. EMT-related markers in TNBC might comprise a signature of a certain sub-group of TNBC[35,36,51], and the EMT programme has an important role in basal breast cancer[52,53]. Thus, in light of our finding that activation of Notch1 can enhance expression of EMT signature genes, including Fibronectin and Slug, we believe that Notch1 may drive TNBC formation under BRCA1-defective conditions by inducing expression of EMT signature genes, thereby promoting the transdifferentiation of luminal cells into the basal-type and at the same time strongly enhancing TNBC progression. This mechanism may also explain why basal-like tumours largely overlap with TNBC in both human and mouse BRCA1-related breast tumours.

Analysis of the human breast cancer database revealed a close correlation between high expression levels of NOTCH1 and TNBC incidence, especially BRCA1-related TNBC. Thus, it is

important to develop an effective therapy for this group of tumours. We recently found that cisplatin, which kills BRCA1-deficient cancers with high efficiency by causing DNA cross-linking[54], also blocks tumour metastasis through inhibition of EMT[41]. Given our finding that NOTCH1 promotes BRCA1-deficient tumour growth by activating ATR–CHK1 and inducing EMT, we inhibited the ATR/CHK1 pathway in combination with a low dose of cisplatin to effectively kill TNBC in our mouse model and PDX model[55]. This approach might avoid the non-specific cytotoxic side effects commonly associated with high doses of cisplatin, thus providing a potent clinical option for TNBC treatment.

PARP1 inhibitors are used for the treatment of BRCA1-deficient cancers, as inhibition of PARP1 generates more single-strand DNA damage, which results in DSBs and consequently leads to a synthetic lethality with BRCA1 deficiency. Our strategy of blocking ATR–CHK1 is likely to target the cell cycle, i.e., it targets different aspects of PARP1. Our data indicate that inhibition of ATR or CHK1 works with cisplatin, which causes DNA inter-strand crosslink, leading to DSBs. As the effect of cisplatin shares some similarities with that of PARP1 inhibitors, we suspect that blocking ATR–CHK1 might also involve PARP1 function, which should be investigated in the near future.

Of note, alteration of many genes involved in some other systems and pathways can accelerate tumourigenesis, including the ubiquitination system, cytoskeleton and cell junction, gene expression regulation, and protein kinase/phosphatase modification-related functions. As Brca1 is an E3 ligase[56], enrichment of ubiquitin-related genes may have a compensatory role in Brca1 deficiency. Cytoskeleton rearrangement is involved in the EMT and tumour metastasis, whereas tight junctions are a specialised structure of epithelial and endothelial cell membranes and have an important role in maintaining most apical junctional complexes[57]. Indeed, components of all enriched pathways among our candidate genes are known to participate in cell proliferation, transformation and metastasis suppression, though their specific role in Brca1-related tumourigenesis remains elusive. Regardless, these data demonstrate that these mutations can activate multiple oncogenic pathways to overcome the lethal block of Brca1 deficiency in mammary epithelial cells to promote the initiation and progression of cancer formation.

In summary, our study revealed that activation of Notch1 has various important consequences, including cell-cycle checkpoint activation and promotion of transdifferentiation by inducing expression of EMT signature genes. By stimulating ATR–CHK1 signalling, activation of the G2/M and S/G2 cell-cycle checkpoints, at least in part, overcomes the lethal block caused by Brca1 deficiency through suppression of the mitotic catastrophe and promotes tumour initiation (Fig. 8g). However, activation of Notch1 also promotes TNBC formation by activating the EMT signalling pathway. The combination of these effects enhances tumourigenesis; conversely, treatment with cisplatin, which inhibits the EMT, and ATR–CHK1 inhibitors that target the cell-cycle checkpoint display good synergy in inhibiting TNBC, thus providing a potent clinical option for this fatal disease.

## Methods

**Mice**. All mouse experiments were performed under the ethical guidelines of the University of Macau (animal protocol number: UMAEC-037-2015). Mice were housed in a Specific-pathogen-free (SPF) facility at 23–25 °C on a 12-h light/dark cycle. The following mouse strains were used in this study. (1) Brca1 conditional knockout ($Brca1^{Co/Co}$) mice, in which deletion of exon 11 of Brca1 is controlled by two mammary tissue-specific Cre transgenes (WAP-Cre or MMTV-Cre)[14]. (2) Two strains with a conditionally expressed SB11 transposase that includes a floxed transcriptional stop cassette to be activated by Cre. (3) Independent transgenic lines of T2onc3 (12740 and 12775)[22], located on chromosome 9 and chromosome 12,

respectively. Because of the several transgenic strains used in this study, the resulting cohorts of mice were of a mixed genetic background, including C57BL/6J, 129SVE and FVB. Mice without the SB transposase or transposon served as a negative control to compare the function of the transposon on tumour formation under Brca1-defective conditions. Animals in all experimental groups were examined twice a week for tumourigenesis. Complete necropsies were performed to assess the primary tumour and metastasis in other organs. Only female mice were used for the experiment, and all the mice were pregnant once at 2–4 months of age to activate expression of WAP-Cre or MMTV-Cre. We named the tumour based on the mouse eartag number and the tumour location information; for example, MK1370-3R means that the tumour was collected from the third right mammary gland of mouse MK1370.

**Histology and IHC**. Routine haematoxylin and eosin staining was performed on 6-μm sections of formalin-fixed, paraffin-embedded (FFPE) tissues. IHC for ERα (1:50, Santa Cruz, sc-542), PR (1:50, Santa Cruz, sc-538), HER2 (1:50, Santa Cruz, sc-284), cleaved caspase-3 (1:500, CST, 9664) and Ki67 (1:200, Abcam, ab16667) was performed to characterise tumour histopathology using a Histostain-Plus IHC kit (Thermo, 859043, Lot: 1954379A) after antigen retrieval (pH 6.0), quenching of endogenous peroxidase and overnight incubation with the primary antibody.

**CIS identification and annotation**. A sequence library of SB transposon insertion sites was constructed using Splinkerette-PCR[58] followed by a second round of PCR with SB Illumina adaptors (Supplementary Data 10). Samples were sequenced using the Illumina HiSeq X-Ten platform with paired-end 150-bp reads. The resulting reads were filtered and trimmed using cutadapt version 1.16 for contaminant sequences, including Illumina adaptor sequences, splinkerette linker sequences and transposon sequences. Reads without a valid adaptor and transposon sequence were excluded from further analysis. Reads <20 bp after trimming were discarded. Clean reads were aligned to the mouse reference genome mm10, allowing 3 mismatches, using Bowtie2 version 2.3.4 followed by TAPDANCE analysis[23]. The aligned locations identified as CISs were automatically annotated based on mouse reference genes (mm10.gtf). CISs located in the gene body plus 3 kb upstream with $p$-values < 0.05 were analysed further.

CIS genes were then annotated with the criteria of gene location plus the upstream 3 kb. Although local hopping of the SB transposon always increased within the chromosome where the transposon concatamer was located, the insertion patterns among the SB strains 12740 or 12775 did not show bias with regard to chromosome, regardless of which Cre (MMTV-Cre or WAP-Cre) was used (Supplementary Fig. 2a). Therefore, we assessed all insertion sites for candidate gene identification, including those in Chr9 or Chr12, which are the original insertion sites for the T2Onc3 transposon in strains 12740 or 12775, respectively. However, to avoid the bias of local hopping, we chose genes identified in more than 5% of the samples in both 12740 and 12775 as strict criteria to filter out fake genes.

CIS genes were considered oncogenic drivers if all insertions were located in a specific region and the CAG promoter of the transposon was in the same orientation as the gene's ORF, which will drive target gene overexpression from the insertion site. Conversely, genes were considered tumour suppressors in the event of an unbiased insertion site and no bias orientation of the CAG promoter, whereby the polyadenylation signal of the transposon most likely disrupts the gene.

**Transcriptome sequencing and analysis**. RNA-seq libraries were constructed using an Illumina TruSeq RNA library Prep Kit and sequenced with the Illumina HiSeq X-Ten platform to generate a minimum of 25 million paired-end 150-bp reads. Sequencing reads were aligned to the mouse reference genome mm10 and processed using HISAT2 version 2.1.0[59]. Differential expression analysis was conducted using DESeq2 version 1.22.2[60]. For exon-level expression, reads from each exon of the target gene were counted followed by normalisation with total reads and length of individual exons to avoid bias due to different exon sizes. PAM50 subtype assignment was conducted using Genefu (2.18.1)[61].

**Functional enrichment analysis**. Gene function annotation enrichment analysis was performed with DAVID Bioinformatics v6.8 using the Gene Ontology and KEGG pathway data sets[62,63]. GSEA (GSEA v3.0)[64] was also used for gene expression difference analysis. Analysis of driver gene molecular interactions among CIS genes was conducted using STRING v11.0 online tools[65].

**Cross-species TNBC correlation analysis**. Gene expression data for human patient samples were downloaded from TCGA[31] and METABRIC[32] databases by using R studio (version 3.5.1) with cgdsr packages. Clinical pathology information was also retrieved. We obtained 1898 and 403 patients from the METABRIC and TCGA databases, respectively. Among them, 37 BRCA1-mutant samples were obtained from the former. For each gene, we ranked the samples based on expression levels. The samples were separated equally into 10 (for all patients) or 6 (for BRCA1-mutant patients) cohorts. Finally, we counted the TNBC proportion of each single cohort to determine the correlation between the gene expression level and TNBC morbidity for each individual gene.

**Western blotting**. Cultured cells were homogenised in RIPA buffer supplemented with phosphatase and protease inhibitor cocktails (Sigma). After measurement and normalisation of the protein concentration, whole-cell lysates were loaded onto polyacrylamide gels for electrophoresis. Proteins were electrotransferred onto PVDF membranes (Bio-Rad), blocked with 5% BSA in TBS-T buffer for 1 h at room temperature and incubated with the primary antibody overnight at 4 °C. The blots were incubated with the corresponding secondary antibody (Cell Signaling, CST), and immunoreactive bands were developed with the ECL substrate (Millipore). The primary antibodies used in this study were as follows: BRCA1 (1:200, Santa Cruz, sc-642), NOTCH1 (1:1000, CST 4380), E-cadherin (1:1000, CST 3195), β-catenin (1:1000, Abcam, ab32572), N-cadherin (1:1000, Abcam, ab76057), Vimentin (1:1000, CST 5741), Fibronectin (1:1000, Abcam, ab2431), Slug (1:1000, CST 9585), β-actin (1:4000, Sigma, A5316), ATM (1:1000, Abcam ab78), pATM (1:1000, Abcam, ab81292), ATR (1:1000, CST 2790), p-ATR-1989 (1:1000, Abcam, ab227851), p-ATR-428 (1:1000, CST 2853), CHK1 (1:200, Santa Cruz, sc-56288), p-CHK1-317 (1:1000,CST 12302) and p-CHK1-345 (1:1000, CST 2348). The secondary antibodies used in this study were as follows: Anti-rabbit IgG, HRP-linked Antibody (1:5000, CST 7074, Lot:28), Anti-mouse IgG and HRP-linked Antibody (1:5000, CST 7076, Lot:32).

**Quantitative RT-PCR**. Total RNA was extracted using TRIzol reagent (Thermo) by following their routine procedure. cDNA was synthesised with the QuantiTect Reverse Transcription kit (Qiagen, 205313). Quantitative PCR was performed using FastStart Universal SYBR Green Master (Rox) mix (Sigma) on a QuantStudio 7 Flex Real-Time PCR System (Thermo). The primer sequences used are listed in Supplementary Data 10.

**Vector construction**. shRNA plasmids were constructed by inserting annealed targeting oligos into the pLKO.1 backbone (Addgene, 8453) after digestion with AgeI and EcoRI (NEB, R0552 and R0101). ICN1 cDNA was cloned from the Puro-iNotch1IC plasmid (Addgene, 75338)[66] for mice and EF.hICN1.CMV.GFP plasmid (Addgene, 17623)[67] for humans and then inserted into the tet-on plasmid (Addgene, 80921)[68]. An HA tag was also added to the N terminus of ICN1. Routine PCR and Sanger sequencing were performed to confirm successful insertion into the vector. The oligonucleotide sequences used for shRNA knock-down are listed in Supplementary Data 10.

**Cell culture**. Mouse primary mammary tumour cells were derived from SB tumours after dissociation with digestion buffer, which contained DMEM/F12 medium (Thermo), 5% FBS, 10 ng/ml EGF (Thermo, 13247-051), 0.5 mg/ml hydrocortisone (Sigma, H0888), 20 ng/ml cholera toxin (Sigma, C-3012), 5 µg/ml insulin (Sigma, 350-020), 300 U/ml collagenase III (Worthington, S4M7602S) and hyaluronidase (Sigma, H3506)[10]. MCF10A cells were cultured in DMEM/F12 medium containing 5% horse serum (Thermo), 20 ng/ml EGF, 0.5 mg/ml hydrocortisone, 100 ng/ml cholera toxin, 10 µg/ml insulin and pen/strep (Thermo). T47D, MCF7, MDA-MB-231, MDA-MB-436, MDA-MB-468, Sum149 and HCC1937 cells were cultured in DMEM supplemented with 10% FBS, glutamine (Thermo), insulin and pen/strep. The human primary breast cancer cell line TM00091 was derived from a PDX model (Jax, TM00091)[55], which carries a BRCA1 mutation with high expression of NOTCH1, by using the same approach used for the mouse primary cells. All cells were routinely tested to exclude mycoplasma contamination.

**Cell-cycle synchronisation**. The double thymidine block approach was applied for cell-cycle synchronisation as previously reported[69]. Briefly, cells were blocked using 2 mM thymidine for 18 h when they reached 25–30% confluence. After 9 h of culture in regular medium, 15 h of thymidine incubation was performed for the second round of blocking. The cells were then released and collected at different time points for cell-cycle flow cytometry analysis.

**MI analysis**. Cells were cultured in 6-well plates. After irradiation, the cells were trypsinised and fixed in 70% cold ethanol for at least 1 h. The cells were permeabilised with 0.25% Triton X-100 for 15 min at room temperature and incubated with 5% BSA/PBS for blocking. The cells were then incubated sequentially with an anti-p-H3 antibody (1:500, Millipore, 06-570), Alexa 488-conjugated secondary antibody (1:1000, Thermo, A-11070, Lot:1431810) and PI. Fluorescence-activated cell sorting (FACS) was performed immediately to analyse the MI[28].

**Fluorescence microscopy**. Cultured cells were fixed with 4% PFA/PBS for 10 min, permeabilised with 0.25% Triton X-100 for 15 min and blocked in 5% BSA/PBS for 1 h at room temperature. For BrdU (1:200, Santa Cruz, sc-32323) staining, DNA was denatured with 2 N HCl for 30 min and neutralised with two round washes of PBS before the blocking step. For EdU staining, the click-iT reaction was carried out after permeabilisation using a Clic-iT EdU Alexa Fluor imaging kit (Thermo, C10339). The primary antibody was incubated overnight at 4 °C. After three washes with PBS, secondary antibodies and DAPI were incubated for 1 h at room temperature. For the quantitative image-based cytometry assay, images were captured and analysed using an In Cell Analyzer 2000 (GE). The γ-H2AX (1:500,

Millipore, 05-636, Lot: 3076468) and 53BP1 (1:200, Santa Cruz, sc-22760, Lot: I1813) immunofluorescence staining are same with MI analysis. Secondary antibodies for this experiments are: F(ab′)2-Goat anti-Rabbit IgG (H + L) Cross-Adsorbed Secondary Antibody, Alexa Fluor 488 (1:1000, Thermo, A-11070, Lot: 1494754), F(ab′)2-Goat anti-Rabbit IgG (H + L) Cross-Adsorbed Secondary Antibody, Alexa Fluor 594 (1:1000, Thermo, A11072, Lot: 1431810), F(ab′)2-Goat anti-Mouse IgG (H + L) Cross-Adsorbed Secondary Antibody, Alexa Fluor 594 (1:1000, Thermo, A-11020, Lot: 1454439), F(ab′)2-Goat anti-Mouse IgG (H + L) Cross-Adsorbed Secondary Antibody and Alexa Fluor 488 (1:1000, Thermo, A-11017, Lot: 1557766)

**Cell viability analysis**. Cell viability was assessed using different approaches, including the MTT assay and the Alamar Blue assay. For the MTT assay, MTT dye (3-[4,5-dimethylthiazol-2-yl]-2,5-diphenyltetrazolium bromide; thiazolyl blue, Sigma-Aldrich) was added (final concentration: 0.5 mg/ml) to cultured cells for 4 h. The formazan crystals were solubilised with DMSO, and absorbance was measured at 570 nm using a plate spectrophotometer. The Alamar Blue assay was used for cell drug treatment analysis. Briefly, culture medium containing 0.02% Alamar Blue was added to cultured cells, and absorbance at 590 nm under 560 nm excitation was measured 2 h of incubation at 37 °C.

**Allograft/xenograft studies**. Tumours or cultured cells were dissociated into single cells and resuspended in 50% Matrigel (Corning, 356234) for inoculation. NSG or nude mice were anaesthetised with tribromoethanol, and a small abdominal incision was made. Mammary fat pads were exposed gently by forceps, and 1 million cells were injected using a microlitre syringe with a 27-gauge needle. The mice were maintained in an SPF facility for further drug administration.

The mice began receiving drug treatment when the tumours reached ~200 mm³. Tumour volume was calculated as $V = (W^2 \times L)/2$[70]. AZD6738 was administered by oral gavage at 10 or 0.5 mg/kg every 3 days together with cisplatin, which was administered by intraperitoneal injection at 1.5 or 5 mg/kg.

**Statistics and reproducibility**. Microsoft Excel, GraphPad Prism (version 6) and R (version 3.5.1) software were used for statistical calculations. Specific statistical tests, sample number and other information are indicated in the main text or figure legends. The t-test and two-sided statistical analysis approach was used to determine the significance of the difference between the different sets of data if there was no specific indication.

All experiments were conducted at least three times independently, and similar results were adopted for further analysis to guarantee reproducibility.

**Reporting summary**. Further information on research design is available in the Nature Research Reporting Summary linked to this article.

## Data availability

The RNA sequence data have been deposited in the Sequence Read Archive (SRA) database under the accession code PRJNA529536. The data from TCGA and METABRIC referenced in the study are available in a public repository from the cBioPortal website (https://www.cbioportal.org/). All the other data supporting the findings of this study can be found within the supplementary files. A reporting summary for this article is available as a Supplementary Information file. Source data for all the plots are provided as a Source Data file.

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

## Acknowledgements

We thank members of the C. Deng and X. Xu laboratories for helpful advice and discussion, the Animal Research Core for providing the animal housing, and the Information and Communication Technology Office (ICTO) for providing the HPC for data processing. This work was supported by the Chair Professor Grant granted to C.D., multi-year research grant (MYRG) 2016-00139-FHS to C.D. and 2018-00186-FHS to K.M. by the University of Macau, Macau SAR, China; the Macao Science and Technology Development Fund (FDCT) Grant 094/2015/A3, 0011/2019/AKP, 0034/2019/AGJ and 0048/2019/A1 to C.D.; 029/2017/A1 and 0101/2018/A3 to X.X., and 111/2017/A to K.M.; and the National Natural Science Foundation of China grant 81672603 to Q.C.

## Author contributions

C.D. and K.M. designed this research; K.M., H.L., J.X., S.C., X.Z., S.M.S. and F.S. performed experiments; H.L., L.W. and S.M.S. performed the histopathological analysis and immunohistochemical staining; J.X., J.B., H.S. and Q.C. assisted with the cell-cycle experiments and data analysis; M.V.V., A.Z., X.L., B.K.R. and J.Z. identified the CISs, analysed the RNA-sequencing data and performed other bioinformatics analyses; K.M., F.S. and P.C. performed the drug treatment experiments; Z.M. and K.H.W. initiated the bioinformatics analysis for identifying CISs; K.T. was responsible for sequencing the SB libraries; K.H.W. and K.M. supervised the bioinformatics analysis; C.D., X.X. and Q.C. supervised the experiments; and K.M. and C.D. wrote the manuscript.

## Competing interests

The authors declare no competing interests.
