## [Peer Review File · Nature Communications]

Reviewers' comments:

Reviewer #1 (Remarks to the Author):

In this manuscript, the authors identify Notch1 activation would favor the development of TNBC in Brca1 deficiency condition and promote EMT. They found that activation of Notch1 could compensate the lethal effect of Brca1 deficiency by activating ATR-CHK1 signaling cascade to bypass cell cycle check point. However, if Notch1-ATR-CHK1 would involve in EMT was not addressed. Also, the link between Notch1 and ATR was not mentioned in this manuscript. Missing some mechanistic details weakens the content of this manuscript. Major revision before publication is recommended.

Below are my concerns:

1. In figure 1e, the authors addressed the effect of transposon on breast cancer spectrum. In this analysis, for BrWSB group, they only assess 20 cases out of 313 (186 + 127). This only represents 6.4% of the study cohort. Due to this small number of cases being assessed, this is hard to make the conclusion stated in lines 108 to 110. Furthermore, HER2 overexpression was not considered in their analysis. I understand the authors wanted to focus on TNBC. However, it is important to show a complete analysis.
2. In figure 2, the authors identified 119 genes and 90 genes from BrWSB and BrMSB respectively. Among them, 40 genes are common, 129 genes (79 + 50; figure 1a) should be the number of distinctive genes. In addition to analyze the 169 candidates together, the authors should break the protein-protein interaction analysis into 119 genes (BrWSB) and 90 genes (BrMSB) to generate two interaction maps. The common clusters among two maps should be the important drivers of Brca1 deficiency tumors. It will be good to further classify the tumors into TNBC and others. This may further drivers for TNBC and other breast cancer types.
3. In figure 3c, the authors may need to explain for what MK1370-3R, MK1058-3L, etc. represent.
4. In figure 3c, in addition to RNA sequencing, the authors should employ western blot or immunohistochemistry to confirm the effect of ICN overexpression in the tumor. Increase the signal on RNA seq does not mean the protein would be enhanced as well. It is important to validate the result.
5. In line 195, the authors may need to clarify the information of MK1097-5R. Was it a tumor without transposon inserted in Notch1 but in other location? Then where was it?
6. In line 200, overexpression of ICN1 should be confirmed on protein level to support the statement.
7. In figure 3g, the authors employed ICN overexpression to demonstrate that activation of Notch signaling could compensate the lethal effect mediated by Brca1 deficiency. In addition, the authors should test the effect of Notch1 ligand on the cell viability as well. This can further support that activation of Notch1 signaling could overcome the lethal effect of Brca2 deficiency.
8. In figure 5, the authors found that Notch1 expression was associated with TNBC. The author should extend their analysis to determine if overexpression of Notch1 would associate with any clinical features of breast cancer, metastasis in particular, and also determine the HR of Notch1 in TNBC.
9. In figure 6a, in addition to EMT, the author should also show other affected gene sets. They may employ a table to show the affected gene set and the enrichment score for each of the gene sets.
10. In figure 6, the authors demonstrated that overexpression of ICN1 (part of Notch1) could promote EMT in Brca1 knockdown cells. It is important to determine if the activation of Notch1 by its ligand will induce similar effect. So that it can conclude overexpression of ICN1 activate Notch1 signaling to induce EMT rather that ICN1 employ other mechanism to induce EMT.
11. The authors found that ATR-CHK1 was important for Notch1 to compromise the lethal effect of Brca1 deficiency and promote EMT. It is important to determine if the activation of ATR-CHK1 cascade would be involved in the promoting EMT.
12. In discussion section, line 444, the authors mentioned that expression of ICN1 represent Notch1 activation. I wonder if it is relevant to clinical feature of TNBC, i.e. overexpression of ICN1,

truncated Notch1, could be detected in human tumor. If not, then in clinical setting, how could we assess Notch1 activity?

13. Since PARP1 inhibitor is also used for Brca1 deficiency patients, the author should discuss if their study would affect the current usage of PARP1 inhibitor.

Reviewer #2 (Remarks to the Author):

Miao and co-workers have investigated the role of non-canonical NOTCH signaling in BRCA1-deficient cells.

The authors performed a sleeping beauty-based insertional mutagenesis screen to identify genetic drivers for tumorigenesis in a background of BRCA1 deficiency. This impressive experiment contained multiple genetic backgrounds and revealed a robust acceleration of tumor formation, with a predominant TNBC phenotype.

Notch was a main hit in the in vivo screens. Analysis of public pt data, Notch overexpression appears to be common in these tumors. Using an overexpression tool to mimick the integration site mutants, the effect of ICN1 on phenotypes of BRCA1-depleted cells are tested. Here the authors do not provide sufficient controls, and over-interpret their data. More importantly, there is no convincing mechanistic insight into how Notch1 activation reverses the lethal phenotypes of BRCA1 loss. The data are mostly associations, and the final experiments confirms that ATR and Chk1 can be used to sensitize cancer cells to cisplatin treatment. There is no link with NOTCH signaling here.

Comments:

- Grammar needs to be checked thoroughly:
- Line 3: 'refractory' is used in oncology to describe whether individual patients respond to treatment. Although I understand what the authors mean, the terms should be used differently.
- Line 7: suppress ->suppresses
- -line 19. This is not the official abbreviation of the BRCA1 gene name
- Line 28: 48% is a number from a specific study. This is not per se valid for the rest of the world. Should be presented as a range from different studies.
- Line 35: We and others only included a reference for 'we', not for 'others'
- Line 37-42: grammar is off, should be rewritten.
- Line 123: word is missing after apical.
- The discussion of line 133-142 is not adding much and is speculative. If anywhere, this should be in the discussion section.
- Line 184: serve-=>serves

- Instead of a string network analysis, GSEA of the (full list) of integration sites would be much better.
- Fig 3. It is unclear to me how the insertion sites when in the sense direction are per se oncogenic. Are insertion sites in exons excluded?
- Fig Suppl 4/3D: how do the authors envision that different shRNAs targeting the same transcript have differential effects? Also, effects on Notch expression at the gene and protein should be assessed using qPCR and WB.
- target gene expression upon dox-mediated induction of ICN1 should be assessed.
- Fig. 4c.D: elevated levels of H2AX are not a read-out of mitotic catastrophe. Live cell imaging or IF should be conducted. Also, the foci should be scored. Current images with low magnification are not informative.
- Fig 4: BRCA1 depletion is far from complete. Same for Chk1.
- Growth assessment is only done in short-term assays, and should be repeated in replating assays. Effects do not seem to be major.
- whereas the in vivo screens are performed in genetically defined models, the in vitro work and

xenograft data comes from inferior models, with partial knockdown.

- Chk1 and ATR inactivation have been demonstrated previously to be lethal in BRCA-defective cells. The fact that shATR and shChk1 leads to cell death in ICN1-overexpressing cells does not mean that this is the mechanism-of-action of ICN1 overexpression.

- To prove this point, the authors should take BRCA1-deficient tumor cells from the screen with ICN1 overexpression, and show that these tumor depend on ICN1 expression, but not when Chk1 is overexpressed.

- The in vivo studies showing that ATR and CHK1 can be therapeutic targets together with cisplatin is in line with many previous studies. There is no evidence that this is linked to Notch signaling.

-

Reviewer #3 (Remarks to the Author):

A. Summary of the key results:

The authors have performed a large Sleeping Beauty transposon screen to identify genes that cooperate with Brca1 loss to promote breast cancer development in the mouse. The authors identified over 100 genes through their screening efforts and focus follow on studies on Notch1 mechanism studies. The authors also performed comparative genomics analyses and test drug combinations based on the findings of their Notch1 studies.

B. Originality and significance: if not novel, please include reference

A quick scan of the literature shows that Notch signaling has been implicated in TNBC previously. However, all of the results from the SB screen are very novel and some of the deeper studies looking at how Notch1 cooperates with Brca1 loss are novel and of high interest.

C. Data & methodology: validity of approach, quality of data, quality of presentation

The approaches used were valid and the data quality is well done.

D. Appropriate use of statistics and treatment of uncertainties

The statistical analyses used appear to be sound.

E. Conclusions: robustness, validity, reliability

Overall the conclusions are robust and valid, except for the items below:

-In figure 7d, why was high Cisplatin+AZD not included as a cohort? High dose cisplatin produced that largest inhibition of tumor growth on its own. I do agree it is notable that low-cisplatin+AZD is better than low cisplatin alone. The sentence beginning on line 420 needs to be restated to be clear on this point. This also needs to be addressed in the discussion, sentence on lines 483-487. Would also be nice to generate data for a high Cisplatin+AZD cohort.

-The conclusion on line 217 notes that ICN functions through suppression of mitotic catastrophe, but the data shown to support this is actually in the next section. Delete or move this conclusion.

F. Suggested improvements: experiments, data for possible revision

-Add densitometry values to western blots in Figure 6d and e

-Add densitometry values to western blots in Supplementary Figure 5

G. References: appropriate credit to previous work?

The references are largely appropriate, though I think the authors need to make it clearer in the text that finding Notch1 as an oncogene in TNBC is not novel and appropriately cite previous studies.

H. Clarity and context: lucidity of abstract/summary, appropriateness of abstract, introduction and

conclusions

-There are a number of poorly structured sentences and grammatical errors. I would suggest having the manuscript reviewed by one or more native English speakers.

Minor points:

-Line 51 says "Identify", should say "Identifying"

-Define and cite TCPA in Supplementary Figure 7

-Figure 7g needs to be referred to in the text somewhere

-Why is the data in figure 7c presented in a different manner than similar data shown in figure 7a?

Responses to Reviewers' comments

➤ Reviewer #1 (Remarks to the Author):

In this manuscript, the authors identify Notch1 activation would favor the development of TNBC in Brca1 deficiency condition and promote EMT. They found that activation of Notch1 could compensate the lethal effect of Brca1 deficiency by activating ATR-CHK1 signaling cascade to bypass cell cycle check point. However, if Notch1-ATR-CHK1 would involve in EMT was not addressed. Also, the link between Notch1 and ATR was not mentioned in this manuscript. Missing some mechanistic details weakens the content of this manuscript. Major revision before publication is recommended.

Thanks a lot for your insightful suggestions, which are critical for this study. We have revised our manuscript according to your suggestions, which will be detailed below.

To study if Notch1-ATR-CHK1 would involve in EMT, we have used CHK1 knockdown cells and ATR knockdown cells for Notch1 induction, followed by checking their EMT status. As shown in the following figure (R-Figure 1), Dox mediated Notch1 induction increases expression of Fibronectin (a commonly used marker for EMT), and knockdown of ATR or CHK1 does not affect this. These data suggest that the function of Notch1 in EMT is independent of ATR or CHK1. We have included this data in lines 409-412, page 17 (Supplementary Figure 9e, f).

R-Figure 1. Western blot analysis of expression of E-Cadherin and Fibronectin during a time course of ICN1 induction upon knockdown of ATR or CHK1.

For the link between Notch1 and ATR, we have now added more data into the text to support the connection, which is summarized below:

- 1). We have conducted a reciprocal IP experiment, which shows ICN1 and ATR can interact with each other (Fig. 4s, also R-Figure 2a).
- 2). p-ATR and p-CHK1 get accumulated in nuclear after induction of ICN1 (Supplementary Fig. 6d, e, also R-Figure 2b, c).
- 3). The similar effect is observed when we use Notch1 ligand Jagged1 to activate the Notch1 signalling (Supplementary Fig. 5f, g, Supplementary Fig. 6c, also R-Figure 2d-f).
- 4). WB shows closely correlation in both MCF10A cells (Fig. 4l) and T47D (Supplementary Fig. 6b) cells when we induce ICN1 overexpression.
- 5). Functionally, ATR knockdown abolishes the rescue effect of Notch1 on cell cycle progression (Fig. 4r).
- 6). To further confirm the correlation between ICN1 and p-ATR in clinical samples, we used two sets of human TNBC patient tissue microarrays to conduct immunohistochemistry

staining (Supplementary Fig. 6f, also R-Figure 2g). The results showed that the p-ATR protein level correlated significantly with the NOTCH1 activation level (Fig. 4t, u, also R-Figure 2h, i)

R-Figure 2. (a) IP analysis indicated that ICN1 can directly bind with ATR. (b, c) IF staining of p-ATR and p-CHK1 at in BRCA1 knockdown or parental MCF10A cells after Dox administration. (d) Activation of ICN1 by Jagged1 suppressed cell death caused by Brca1 acute knockdown in MCF10A cells. (e) Quantification analysis by MTT assays regarding the rescue effect of Notch1's ligand on BRCA1 deficiency. Analysis on the 3rd day after Dox and/or lentivirus with shRNA-BRCA1 induction. (f) Western blotting analysis of cell cycle checkpoint proteins and EMT marker proteins at different time points after Jagged1 administration in MCF10A cells. (g) Flowchart illustrating the workflow of human tissue microarray IHC staining. (h, i) Immunohistochemistry staining of a human TNBC patient tissue microarray for target proteins. The scatter plots indicate a positive correlation between ICN1 and p-ATR (n=90 for h, n=72 for i).

Below are my concerns:

1. In figure 1e, the authors addressed the effect of transposon on breast cancer spectrum. In this analysis, for BrWSB group, they only assess 20 cases out of 313 (186 + 127). This only represents 6.4% of the study cohort. Due to this small number of cases being assessed, this is hard to make the conclusion stated in lines 108 to 110. Furthermore, HER2 overexpression was not considered in their analysis. I understand the authors wanted to focus on TNBC. However, it is important to show a complete analysis.

We have now analysed a total of 149 tumours from BrWSB mice and detected 85 (57%) TNBCs (Fig. 1e, also shown as R-Figure 3a below). There are 12 (8.1%) Her2+ cases. Status of expression of these genes is indicated below. Meanwhile, we also analysed 24 BrM control tumours and 48.5% (11/24) were TNBC (Supplementary Fig. 1e, also R-Figure 3b).

R-Figure 3. TNBC incidence among different groups. (a) TNBC incidence in BrW and BrWSB groups. (b) TNBC incidence in BrM groups.

2. In figure 2, the authors identified 119 genes and 90 genes from BrWSB and BrMSB respectively. Among them, 40 genes are common, 129 genes (79 + 50; figure 1a) should be the number of distinctive genes. In addition to analyze the 169 candidates together, the authors should break the protein-protein interaction analysis into 119 genes (BrWSB) and 90 genes (BrMSB) to generate two interaction maps. The common clusters among two maps should be the important drivers of Brca1 deficiency tumours. It will be good to further classify the tumours into TNBC and others. This may further drivers for TNBC and other breast cancer types.

We have used the 119 and 90 gene lists to do the protein-protein interaction analysis. Although the shape of two groups shares some similarities, we do not find common clusters, (R-Figure 4a, b), perhaps due to the fact that there are more different genes than common ones in these two groups. Because the 119 and 90 genes were all identified through functional analysis, we hypothesized that other genes, in addition to the 40 overlapping genes, might also be involved in accelerating tumourigenesis, Therefore, we conducted protein-protein interaction analysis using all 169 candidate genes and observed several clusters (as shown in manuscript Figure 2c).

R-Figure 4. (a, b) Protein-protein interaction analysis of 90 candidate genes from BrMSB group (a) and 119 candidate genes from BrWSB group (b). (c) Putative SB driver genes for both TNBC and non-TNBC tumours.

Now, we have 85 TNBCs and 64 Non-TNBCs. We have started to check the putative SB driver genes in both groups. Our preliminary analysis indicated (as shown in R-Figure 4c) that 42 genes, including Notch1, Atp6v1d, Gpc5 et al. appeared at higher frequencies in TNBCs than in the Non-TNBCs. Whereas 27 genes, including Fbxl17, Peak1os, Phip, etc. appeared other way around. We feel this data is very preliminary to show, and, therefore, we choose to focus on the potential relationship Notch1 and TNBC in the remaining of our study. In our future study, the potential role of some other genes with TNBC will be investigated.

3. In figure 3c, the authors may need to explain for what MK1370-3R, MK1058-3L, etc. represent.

We named the tumour with the mouse eartag number and the tumour location information; for example, MK1370-3R means the tumour was collected from the third right mammary gland of mouse with eartag number: MK1370. We have described it in the methods part.

4. In figure 3c, in addition to RNA sequencing, the authors should employ western blot or immunohistochemistry to confirm the effect of ICN overexpression in the tumour. Increase the signal on RNA seq does not mean the protein would be enhanced as well. It is important to validate the result.

Thanks for your suggestions, we have done Western blot and the data detects ICN1 overexpression in the mouse SB tumours (Fig. 3d, also R-Figure 5).

R-Figure 5. Activated Notch1 protein expression analysis based on western blotting compared with non-Notch1-driven tumours.

5. In line 195, the authors may need to clarify the information of MK1097-5R. Was it a tumour without transposon inserted in Notch1 but in other location? Then where was it?

MK1097-5R is a tumour without transposon insertion in Notch1 locus, instead, our CIS analysis suggests this tumour might be driven by Met activation.

6. In line 200, overexpression of ICN1 should be confirmed on protein level to support the statement.

We have conducted the Western blot, and the result indicates the protein level of Notch1 (Fig. 3e, also R-Figure 6). In addition to the functional assay, we could conclude that transposon insertion-induced overexpression of ICN1 is the driver for MK1370-3R tumours but not for MK1097-5R tumour, which may be driven by other factors.

R-Figure 6. Western blot analysis of Notch1-driven tumours (MK1370-3R) and non-Notch1-driven tumours (MK1097-5R) after shRNA knockdown.

7. In figure 3g, the authors employed ICN overexpression to demonstrate that activation of Notch signaling could compensate the lethal effect mediated by Brca1 deficiency. In addition, the authors should test the effect of Notch1 ligand on the cell viability as well. This can further support that activation of Notch1 signaling could overcome the lethal effect of Brca2 deficiency.

We have used Jagged1, which is the ligand of Notch1, to activate the ICN1, and then we checked the rescue effect for lethality associated with BRCA1 deficiency. The results show that Jagged1 could activate ICN and rescue the lethality (Supplementary Fig. 5e, f, also R-Figure 7a, b)

R-Figure 7. (a) Activation of ICN1 by Jagged1 suppressed cell death caused by Brca1 acute knockdown in MCF10A cells. (b) Quantification analysis by MTT assays regarding the rescue effect of Notch1's ligand on BRCA1 deficiency. Analysis on the 3rd day after Dox and/or lentivirus with shRNA-BRCA1 induction.

8. In figure 5, the authors found that Notch1 expression was associated with TNBC. The author should extend their analysis to determine if overexpression of Notch1 would associate with any clinical features of breast cancer, metastasis in particular, and also determine the HR of Notch1 in TNBC.

Our analysis of TCGA (n=505) and METABRIC (n=1898) databases indicated higher expression of NOTCH1 is correlated with TNBC at both RNA level (Fig. 5d), and protein level (Supplementary Figure 8a, b, also R-Figure 8a, b). Because both the TCGA and METABRIC database (R-Figure 8c, d) showed similar result, we combined RNA level data from both data into one panel and show it in figure 5d.

R-Figure 8. Correlation between Notch1 protein (a, b) or RNA level (c, d) level and tumour subtype. Correlation analysis between NOTCH1 expression and TNBC incidence for TCGA database (c) and METABRIC database (d).

Recently, there is a paper reported an accuracy approach to detect the mutational signature associated with HR deficiency¹. We have used this method to classify the TCGA breast cancer patients into different subtypes. Then detected the expression level of NOTCH1 in different cohorts. The results showed tumours with HR signature has relatively high level of Notch1 gene (Supplementary Fig. 6i, also R-Figure 9a).

We have also analysed the relationship between metastasis and Notch1 expression using TCGA breast cancer dataset, and the data indicates that metastasis didn't significant correlate with Notch1 expression (R-Figure 9b).

R-Figure 9. (a) NOTCH1 expression level analysis in different patient cohort to indicate NOTCH1 correlated with homologues recombination features. (b) NOTCH1 expression in metastasis patient and non-metastasis tumours.

9. In figure 6a, in addition to EMT, the author should also show other affected gene sets. They may employ a table to show the affected gene set and the enrichment score for each of the gene sets.

We have listed all the enriched gene sets in Supplementary Table 8, and modified the text accordingly.

10. In figure 6, the authors demonstrated that overexpression of ICN1 (part of Notch1) could promote EMT in Brca1 knockdown cells. It is important to determine if the activation of Notch1 by its ligand will induce similar effect. So that it can conclude overexpression of ICN1 activate Notch1 signaling to induce EMT rather that ICN1 employ other mechanism to induce EMT.

We have conducted the experiment, and the data shows Notch1's ligand also induces expression of genes involved in the EMT in Brca1 knockdown cells (Supplementary Fig. 6c, also R-Figure 10).

R-Figure 10. Western blotting analysis of cell cycle checkpoint proteins and EMT marker proteins at different time points after Jagged1 administration in MCF10A cells.

11. The authors found that ATR-CHK1 was important for Notch1 to compromise the lethal effect of Brca1 deficiency and promote EMT. It is important to determine if the activation of ATR-CHK1 cascade would be involved in the promoting EMT.

In our study, we found ATR-CHK1 was important for Notch1 to compromise the lethal effect of Brca1 deficiency. We did not mean that ATR-CHK1 contribute to EMT. (We are sorry if we did not make it clear and caused the confusion)

As described earlier (R-Figure 1), Dox mediated Notch1 induction increases expression of Fibronectin, and KD ATR or CHK1 does not affect this, suggesting that the function of Notch1 in EMT is independent of ATR or CHK1.

12. In discussion section, line 444, the authors mentioned that expression of ICN1 represent Notch1 activation. I wonder if it is relevant to clinical feature of TNBC, i.e. overexpression of ICN1, truncated Notch1, could be detected in human tumour. If not, then in clinical setting, how could we assess Notch1 activity?

We have analysed human breast cancer database and found that overexpression of Notch 1 is associated with TNBC both in RNA level (Fig. 5d) and protein level (Supplementary Figure 8a, b, also R-Figure 8a, b).

Mutational analysis of Notch genes also illustrated that aberrant Notch activation happened through truncations in the PEST domain to elongate protein half-life or missense mutations led to ligand-independent processing^{2,3}.

Reference:

Wang, N.J. et al. Loss-of-function mutations in Notch receptors in cutaneous and lung squamous cell carcinoma. *Proc Natl Acad Sci U S A* 108, 17761-6 (2011);

Bhanushali, A.A. et al. Mutations in the HD and PEST domain of Notch-1 receptor in T-cell acute lymphoblastic leukemia: report of novel mutations from Indian population. *Oncol Res* 19, 99-104 (2010).

In clinic setting, there are few approaches to detect Notch1 activation. Firstly, higher expression of Notch1 leads to hyperactivation of Notch1 signaling, which can be detected by RNA expression. Secondly, quantification of activated NOTCH1 with western blot or immunohistochemistry staining is the most direct indicator. Specifically, aa 1755-1767 (intracellular) (VLLSRKRRRQHGQC) only exposed after gamma secretase cleavage and is not accessible in the uncleaved form. Moreover, cell membrane to nuclear translocation of NOTCH1 also indicates ICN1 get released/activated from membrane. In summary, RNA expression level, ICN1 protein expression level, and the subcellular localization of NOTCH1 all could serve as the marker for detection/indication of Notch1 activity.

13. Since PARP1 inhibitor is also used for Brca1 deficiency patients, the author should discuss if their study would affect the current usage of PARP1 inhibitor.

PARP1 inhibitor is used for treatment of BRCA1 deficient cancers as inhibition of PARP1 would generate more single strand DNA damage, consequently, more double strand breaks (DSBs). Thus, PARP1i targets BRCA1 mutant cells for its deficiency in DSBs and therefore generates a synthetic lethality. Our strategy of blocking NOTCH1-ATR-CHEK1 is more likely to target cell cycle, i.e. it targets different aspect with PARP1. Our data indicate that inhibition of ATR or CHEK1 works with Cisplatin, which causes DNA inter-strand crosslink, leading to DSBs. As the effect of cisplatin shares some similarities with that of PARP1i, we suspect that block NOTCH1-ATR-CHEK1 might also work with PARP1i. we also added this into discussion section (Lines 540-547).

➤ **Reviewer #2 (Remarks to the Author):**

Miao and co-workers have investigated the role of non-canonical NOTCH signaling in BRCA1-deficient cells.

The authors performed a sleeping beauty-based insertional mutagenesis screen to identify genetic drivers for tumorigenesis in a background of BRCA1 deficiency. This impressive experiment contained multiple genetic backgrounds and revealed a robust acceleration of tumour formation, with a predominant TNBC phenotype.

Notch was a main hit in the in vivo screens. Analysis of public pt data, Notch overexpression appears to be common in these tumours. Using an overexpression tool to mimick the integration site mutants, the effect of ICN1 on phenotypes of BRCA1-depleted cells are tested. Here the authors do not provide sufficient controls, and over-interpret their data. More importantly, there is no convincing mechanistic insight into how Notch1 activation reverses the lethal phenotypes of BRCA1 loss. The data are mostly associations, and the final experiments confirms that ATR and Chk1 can be used to sensitize cancer cells to cisplatin treatment. There is no link with NOTCH signaling here.

Thanks a lot for your insightful suggestions, which are critical for this study. We have revised our manuscript according to your suggestions, which are detailed below.

Comments:

- Grammar needs to be checked thoroughly:

We have sent the manuscript out to an editing service to correct any possible grammar errors (R-Figure 11).

R-Figure11. Editing Certificate from the AJE.

- Line 3 (now line 24): ‘refractory’ is used in oncology to describe whether individual patients respond to treatment. Although I understand what the authors mean, the terms should be used differently.

We have replaced the “refractory” with “most difficult for the anticancer drug treatment”.

- Line 7 (now line 29): suppress ->suppresses

We have corrected it.

-line 19 (now line 40): This is not the official abbreviation of the BRCA1 gene name

We have changed it to “Breast cancer gene 1 (BRCA1)”.

- Line 28 (now line 47): 48% is a number from a specific study. This is not per se valid for the rest of the world. Should be presented as a range from different studies.

We have checked more publications, and have modified the text as following:

“BRCA1-defective breast cancers are usually high grade and have poor prognoses. 48% to 66% percent of BRCA1 mutation carriers develop triple-negative breast cancer (TNBC); this rate is much higher than that of non-carriers (around 20%)⁴⁻⁷.”

- Line 35 (now line 54): We and others only included a reference for ‘we’, not for ‘others’

We have added references for others.

- Line 37-42 (now line 58-63): grammar is off, should be rewritten.

We have modified the sentences.

- Line 123 (now line 146): word is missing after “apical”.

We have modified the sentence.

- The discussion of line 133-142 is not adding much and is speculative. If anywhere, this should be in the discussion section.

Thanks a lot. We have moved this part into discussion section (lines 551-560).

- Line 184 (now line 204): serve=>serves

We have corrected it.

- Instead of a string network analysis, GSEA of the (full list) of integration sites would be much better.

Thanks for your kindness suggestion, we have tried to use GSEA analysis of the full list of integration sites, but enrichment is not strong. As shown in the R-Table 1, no any pathway reaches a significant p value <0.05 (although three pathways showed an adjusted p value <0.05, but their other parameters are not significant). Therefore, we have also conducted KEGG and GO analysis and the data showed there are some functional items got enriched including ubiquitin mediate proteolysis, cell adhesion related function, and so on (Supplementary Tables 5, 6). The protein-protein interaction analysis for all 169 candidate genes highlighted several clusters, including the ubiquitination system, cytoskeleton and cell

junction, gene expression regulation, and protein kinase/phosphatase modification-related functions. Therefore, we would keep the String for protein-protein analysis in our manuscript.

R-Table 1. Gene set enrichment analysis of all candidate genes.

ID	enrichmentScore	NES	pvalue	p.adjust	qvalues	rank	leading edge	core enrichment
REACTOME DISEASE	15	0.56	1.59	0.02	0.52	0.52	11	tags=27%, list=7%, signal=27%
DAVICIONI MOLECULAR ARMS VS ERMS UP	10	0.61	1.59	0.02	0.52	0.52	18	tags=40%, list=11%, signal=38%
REACTOME DISEASES OF SIGNAL TRANSDUCTION	10	0.62	1.60	0.02	0.52	0.52	11	tags=30%, list=7%, signal=30%
BLALOCK ALZHEIMERS DISEASE UP	31	0.45	1.46	0.05	0.58	0.58	52	tags=48%, list=31%, signal=41%
ONDER CDH1 TARGETS 2 DN	12	0.53	1.45	0.06	0.58	0.58	43	tags=50%, list=25%, signal=40%
BENPORATH SOX2 TARGETS	10	0.56	1.44	0.07	0.58	0.58	49	tags=60%, list=29%, signal=45%
REACTOME GENERIC TRANSCRIPTION PATHWAY	13	0.51	1.42	0.08	0.58	0.58	41	tags=46%, list=24%, signal=38%
BENPORATH NANOG TARGETS	16	0.50	1.44	0.08	0.58	0.58	36	tags=44%, list=21%, signal=38%
REACTOME GENE EXPRESSION TRANSCRIPTION	16	0.50	1.44	0.08	0.58	0.58	41	tags=44%, list=24%, signal=37%
ACEVEDO LIVER CANCER UP	11	0.51	1.36	0.12	0.69	0.69	11	tags=27%, list=7%, signal=27%
DUTERTRE ESTRADIOL RESPONSE 24HR DN	10	-0.38	-1.38	0.13	0.69	0.69	109	tags=100%, list=64%, signal=38%
KOINUMA TARGETS OF SMAD2 OR SMAD3	11	0.50	1.32	0.14	0.69	0.69	36	tags=45%, list=21%, signal=38%
REACTOME POST TRANSLATIONAL PROTEIN MODIFICATION	18	0.45	1.33	0.14	0.69	0.69	12	tags=22%, list=7%, signal=23%
SCHAEFFER PROSTATE DEVELOPMENT 6HR DN	11	0.49	1.29	0.16	0.73	0.73	49	tags=55%, list=29%, signal=41%

- Fig 3. It is unclear to me how the insertion sites when in the sense direction are per se oncogenic. Are insertion sites in exons excluded?

The Insertion sites in exons are included. Usually, if SB transposon inserted into regulatory region, such as promoter, or exon region, will cause disruptive regardless of the orientation of the SB transposon. Even inserted into the intron region, it will disrupt the genes because of the splice acceptor and polyA. But because of there is CAG promoter in transposon, therefore it will initiate overexpression from the following exon. Then if the orientation is same with the harboured gene. It will overexpress the gene.

Here are the details about the SB transposon.

The T2/Onc transposon is specifically designed to induce either gain-of-function (R-Figure 12 B-a, b) or loss-of-function (R-Figure 12 B-c, d) mutations when inserted in or near a gene based on its genetic cargo. The CAG promoter with artificial exon and splice donor was included so downstream exons could be ectopically overexpressed as a consequence of fusion with transcripts initiated by the CAG and splicing from the T2/Onc3 splice donor. The T2/Onc3 vectors also include splice acceptors in both orientations and a bidirectional poly-adenylation signal, to terminate transcripts that splice into the vector after insertion within an intron of a gene. In this case, many tumour suppressor genes (TSGs) will be inactivated as a consequence of SB insertional mutagenesis in various screens.

R-Figure 12. A, Structure of the T2/Onc3 transposon; B. Mechanisms of transposon induced mutations. T2/Onc3 transposons can function to induce gain-of-function mutations in oncogenes (a and b) or to trap the promoter of a tumour suppressor gene in either orientation (c and d); C. Rosa26 SB transposase knock-in alleles.

CIS genes were considered oncogenic drivers if all insertions were located in a specific region and the CAG promoter of transposons is in the same orientation as the gene's ORF, which will drive target gene overexpression from the insertion site. Conversely, genes were considered TSGs with an unbiased insertion site and lacking bias orientation of the CAG promoter, which most likely disrupt the harboured genes.

- Fig Suppl 4/3D: how do the authors envision that different shRNAs targeting the same transcript have differential effects? Also, effects on Notch expression at the gene and protein should be assessed using qPCR and WB.

For Notch1, different shRNAs targeting different regions might give different results because of a special feature of this gene.

As indicated by Supplementary Fig. 4a, b, also R-Figure 13a and b that endogenous Notch1 activation is depended on the ligand binding. When SB transposon is inserted into the cleavage domain (Exon 25-30), the transcript RNA doesn't contain the N-terminal of Notch1 and it shows ligand independent manner. So we have designed one shRNA that targets the N terminal domain, which only knockdown the endogenous Notch1, and another shRNA that targets the ICN domain to knockdown both endogenous Notch1 and transposon driven Notch1.

R-Figure 13. (a) Illustration of the endogenous Notch1 expression, which activation shows ligand dependent manner. And SB transposon driven Notch1 expression, which activation shows ligand independent manner. (b) shRNAs were designed against the N-terminus of endogenous Notch1 or SB transposon derived ICN1.

Our WB result revealed that the design indeed targets different format of Notch1 (Fig. 3e, also R-Figure 14).

R-Figure 14. Western blot analysis of Notch1-driven tumours (MK1370-3R) and non-Notch1-driven tumours (MK1097-5R) after shRNA knockdown.

- target gene expression upon dox-mediated induction of ICN1 should be assessed.

We detected the Notch1 target genes, Hes1 and Hey1, the data is shown in Supplementary Fig. 5c (R-Figure 15).

R-Figure 15. Quantitative analysis of Notch1 targeting gene expression level after Dox administration.

- Fig. 4c.D: elevated levels of H2AX are not a read-out of mitotic catastrophe. Live cell imaging or IF should be conducted. Also, the foci should be scored. Current images with low magnification are not informative.

We have counted the IF foci for γ H2AX, and the data show that shRNA-BRCA1 induces significant more IF foci, which is reduced upon Dox induced expression of NOTCH1 (Fig. 4d also R-Figure 16a). Besides, we also checked the cell imaging and IF of 53BP1 and detected similar data (Supplementary Fig. 5d, e, also R-Figure 16b, c).

R-Figure 16. (a) IF staining and foci counting of γ -H2AX to indicate DNA damage at 48 hours after BRCA1 acute knockdown with or without ICN1 overexpression. (b) IF staining of 53BP1 to indicate DNA damage at 48 hours after BRCA1 acute knockdown with or without ICN1 overexpression. (c) Intensity quantification of 53BP1 to indicate DNA damage at 48 hours after BRCA1 acute knockdown with or without ICN1 overexpression.

- Fig 4: BRCA1 depletion is far from complete. Same for Chk1.

Because complete knockout of these genes is lethal, therefore, we could only use knockdown approach. The references are listed as following:

Reference for Brca1 lead lethal:

- > The tumor suppressor gene Brca1 is required for embryonic cellular proliferation in the mouse⁸
- > A targeted disruption of the murine Brca1 gene causes gamma-irradiation hypersensitivity and genetic instability⁹

Reference for Chk1 lead lethal:

- > Chk1 is an essential kinase that is regulated by Atr and required for the G(2)/M DNA damage checkpoint¹⁰
- > Aberrant cell cycle checkpoint function and early embryonic¹¹

In some special situations, Brca1 complete knockout cell lines could be generated, i.e. loss of p53, and/or some other unknown factors overcome lethality. In the present study, we also showed that knockout Notch1 could also partially rescue the lethality of Brca1 mutant ES cells (Fig. 3i). But for studying acute effect of Brca1 and Chk1, knockdown, rather than knockout out, needs to be used.

- Growth assessment is only done in short-term assays, and should be repeated in replating assays. Effects do not seem to be major.

Thanks for your suggestion, we have replated the cells 4 times and checked cell number during 2 weeks period. The data showing in Supplementary Fig. 5h (R-Figure 17).

R-Figure 17. Growth rate of MCF10A cells under ICN1 overexpression and/or Brca1 knockdown conditions. Cell number was measured every three days, followed by re-plating the cells. Three repeats were used to determine the SD at each point.

- whereas the in vivo screens are performed in genetically defined models, the in vitro work and xenograft data comes from inferior models, with partial knockdown.

We do agree that in vivo screen is better. But to work out mechanisms, people usually use cell culture system. As explained earlier, because of lethality associated with BRCA1 deficiency, we cannot obtain cell lines that completely loss BRCA1. Therefore, knockdown in these cells were used.

Beside using MCF10A cells, we also checked T47D cells and the similar results have been observed.

As shown in Fig. 3, we have demonstrated that Notch1 is critical for the Brca1 mutated tumour growth.

- Chk1 and ATR inactivation have been demonstrated previously to be lethal in BRCA-defective cells. The fact that shATR and shChk1 leads to cell death in ICN1-overexpressing cells does not mean that this is the mechanism-of-action of ICN1 overexpression.

To prove this point, the authors should take BRCA1-deficient tumour cells from the screen with ICN1 overexpression, and show that these tumour depend on ICN1 expression, but not when Chk1 is overexpressed.

We have carefully considered your suggestion. To conduct this experiment, we need to use BRCA1-deficient tumour cells from the screen with ICN1 overexpression. This is easy part, as we have already established multiple cell lines from this type of tumour. However, we also need to overexpress Chk1 in these cells. This part is very difficulty to do. Indeed, our past experience indicated that over expression of Chk1 would prevent cells from proliferation. While we did not study it in details, Dr. O'Connell et al. indicated the over expression of Chk1 causes G2 arrest in the transfected cells¹² (R-Figure 18). Thus, we are limited to conduct this experiment.

[REDACTED]

R-Figure 18. Overexpression of Chk1 blocks cell growth. *chk1* was expressed from the *nmt1* promoter in wild-type cells, and cells carrying an empty vector were used as a control. Induction of *chk1* expression revealed by Western blot was shown in the lower panel. *chk1*. (adapted from Dr. O'Connell et al. The EMBO Journal Vol.16 No.3 pp.545–554, 1997.)¹²

We agree that it is important to uncover the mechanism-of-action of ICN1 and demonstrate clearly regarding its impact on ATR-CHK1. Therefore we have conducted several new experiments to provide more supporting data, which are briefly summarized below:

- 1). We have conducted a reciprocal IP experiment, which shows ICN1 and ATR interact with each other (Fig. 4s).
- 2). p-ATR and p-CHK1 get accumulated in nuclear after induction of ICN1 (Supplementary Fig. 6d, e).
- 3). The similar effect is observed when we use Notch1 ligand Jagged1 to activate the Notch1 signalling (Supplementary Fig. 5f, g, and Supplementary Fig. 6c).
- 4). WB shows closely correlation in both MCF10A cells (Fig. 4l) and T47D (Supplementary Fig. 6b) cells when we induce ICN1 overexpression.
- 5). Functionally, ATR knockdown abolishes the rescue effect of Notch1 on cell cycle progression (Fig. 4r).
- 6). It is well established that ATR phosphorylates CHK1. Consistently, our data indicated that inhibition of ATR reduces pCHK1 (Supplementary Fig. 6a).
- 7). To further confirm the correlation between ICN1 and p-ATR in clinical samples, we used two sets of human TNBC patient tissue microarrays to conduct immunohistochemistry

staining (Supplementary Fig. 6f). The results showed that the p-ATR protein level correlated significantly with the NOTCH1 activation level (Fig. 4t, u)

- The *in vivo* studies showing that ATR and CHK1 can be therapeutic targets together with cisplatin is in line with many previous studies. There is no evidence that this is linked to Notch signaling.

Our main purpose of the *in vivo* studies showing that inhibition of ATR or CHK1 with cisplatin is quite potent to TNBC in our model systems is to provide clinical application for the future development of a therapeutic option of TNBC, which is most difficult to treat. This is based on our observation that a high expression level of Notch1 correlates closely with TNBC incidence (Figure 5), and NOTCH1 interacts with ATR and activates ATR-CHK1 signalling to compensate for the DNA damage deficiency and benefit tumourigenesis, and meanwhile, cisplatin causes DNA cross-link, leading to DNA double strand breaks (DSBs). Therefore we investigated whether the ATR or CHK1 inhibitor combined with cisplatin could have a synthetic lethal effect on TNBC samples. Our data illustrated that inhibition of ATR or CHK1 (which is a target of ATR) in combination with cisplatin works quite well (Figure 7). We have revised the related description in lines 432-434 to indicate this.

Our further study revealed that inhibition of TM00091 PDX tumours, which carry two BRCA1 point mutations (Q356R and C61G) with NOTCH1 hyperactivation (Supplementary Figure 10a), also works quite well. In contrast, inhibition of ATM, which is not affected by NOTCH1, has no such an inhibition effect on TNBCs (Figure 7c). These observations, together with our earlier findings that NOTCH1 interacts and activates ATR-CHK1 (as summarized for your above questions), only demonstrate the link between Notch1 and ATR/CHK1, but also illustrate a potential clinic application for the treatment of TNBC.

➤ **Reviewer #3 (Remarks to the Author):**

A. Summary of the key results:

The authors have performed a large Sleeping Beauty transposon screen to identify genes that cooperate with Brca1 loss to promote breast cancer development in the mouse. The authors identified over 100 genes through their screening efforts and focus follow on studies on Notch1 mechanism studies. The authors also performed comparative genomics analyses and test drug combinations based on the findings of their Notch1 studies.

B. Originality and significance: if not novel, please include reference

A quick scan of the literature shows that Notch signaling has been implicated in TNBC previously. However, all of the results from the SB screen are very novel and some of the deeper studies looking at how Notch1 cooperates with Brca1 loss are novel and of high interest.

C. Data & methodology: validity of approach, quality of data, quality of presentation

The approaches used were valid and the data quality is well done.

D. Appropriate use of statistics and treatment of uncertainties

The statistical analyses used appear to be sound.

E. Conclusions: robustness, validity, reliability

Thanks a lot for your high evaluation and insight suggestions, which are critical for this study. We have revised our manuscript according to your suggestions, which are detailed below.

Overall the conclusions are robust and valid, except for the items below:

-In figure 7d, why was high Cisplatin+AZD not included as a cohort? High dose cisplatin produced that largest inhibition of tumour growth on its own. I do agree it is notable that low-cisplatin+AZD is better than low cisplatin alone. The sentence beginning on line 420 needs to be restated to be clear on this point. This also needs to be addressed in the discussion, sentence on lines 483-487. Would also be nice to generate data for a high Cisplatin+AZD cohort.

We have conducted the high Cisplatin +AZD6738 combine treatment. The results showed both high and low dosages of AZD6738 could significantly increase the killing effect when a high dosage of cisplatin was administered (R-Figure 19a). However, high dosage of cisplatin dramatically decreased the mouse body weight (Supplementary 10b, also R-Figure 19b), indicating a non-specific toxic effect of the treatment. This is our major concern, therefore, we just kept the low dosage of cisplatin treatment in the manuscript. We have addressed this in the “Discussion” (line 536-539).

R-Figure 19 (a) In vivo effect of high dosage of cisplatin combined with ATR inhibitor (AZD6738) on tumours in the PDX model (TM00091), which carries the BRCA1 mutation and highly expresses NOTCH1. Six mice were used for each treatment group. (b) Body weight analysis of high-dosage and low-dosage of cisplatin treatment of the PDX model (TM00091)

-The conclusion on line 217 notes that ICN functions through suppression of mitotic catastrophe, but the data shown to support this is actually in the next section. Delete or move this conclusion.

We have modified the sentence (now on line 239) “.....which may occur by suppressing the lethality effect caused by Brca1 deficiency.”.

F. Suggested improvements: experiments, data for possible revision

-Add densitometry values to western blots in Figure 6d and e

-Add densitometry values to western blots in Supplementary Figure 5

We have added the densitometry values into both figure 6d, e and Supplementary Figure 5 (now Supplementary Figure 6).

G. References: appropriate credit to previous work?

The references are largely appropriate, though I think the authors need to make it clearer in the text that finding Notch1 as an oncogene in TNBC is not novel and appropriately cite previous studies.

Thanks a lot for your kindness suggestion. There are two issues we need discuss. The first one is that Notch1 seems has a dual function in cancer, as a oncogene or tumor suppressor under different situation. Therefore we revised two places to discuss these. More references are also cited accordingly.

In lines 202-205: we added “Although the role of Notch1 in cancer formation has been well established, it is somewhat surprising that it could act as an oncogene or a tumour suppressor under different context^{13,14}. Thus, we decided to focus on Notch1 to illustrate the mechanism underlying its oncogenic function”.

In lines 512-516, we added the following sentences: “Previous studies have revealed that activation of NOTCH1 is correlated with cancer stem cell maintenance and expansion¹⁵, and that the high expression level of NOTCH1 may serve as a prognostic marker in patients with TNBC¹⁶, yet the underlying mechanism remains elusive. Our study, at the first time from functional screening, demonstrates that the activation of Notch1 promotes TNBC formation.”.

H. Clarity and context: lucidity of abstract/summary, appropriateness of abstract, introduction and conclusions

-There are a number of poorly structured sentences and grammatical errors. I would suggest having the manuscript reviewed by one or more native English speakers.

We have send out our manuscript for editing. We believe the wirting is improved.

Minor points:

-Line 51 (now line 72) says “Identify”, should say “Identifying”

We have modified the word.

-Define and cite TCPA in Supplementary Figure 7

We have cited the paper and modify the text accordingly.

-Figure 7g needs to be referred to in the text somewhere

We cited Figure 7g in the discussion part.

-Why is the data in figure 7c presented in a different manner than similar data shown in figure 7a?

We have conducted the combined drug treatment for several different cell lines, including MDA-MB-436, MDA-MB-468, Sum149, etc. Figure 7a just shows the representative data to indicate the synergistic effect between ATR inhibitor and cisplatin. But we have simplified the data format to avoid the figure redundant, which has been showed in Figure 7c.

References

1. Gulhan, D.C., Lee, J.J., Melloni, G.E.M., Cortes-Ciriano, I. & Park, P.J. Detecting the mutational signature of homologous recombination deficiency in clinical samples. *Nat Genet* **51**, 912-919 (2019).
2. Wang, N.J. *et al.* Loss-of-function mutations in Notch receptors in cutaneous and lung squamous cell carcinoma. *Proc Natl Acad Sci U S A* **108**, 17761-6 (2011).
3. Bhanushali, A.A. *et al.* Mutations in the HD and PEST domain of Notch-1 receptor in T-cell acute lymphoblastic leukemia: report of novel mutations from Indian population. *Oncol Res* **19**, 99-104 (2010).
4. Lee, E. *et al.* Characteristics of triple-negative breast cancer in patients with a BRCA1 mutation: results from a population-based study of young women. *J Clin Oncol* **29**, 4373-80 (2011).
5. Spurdle, A.B. *et al.* Refined histopathological predictors of BRCA1 and BRCA2 mutation status: a large-scale analysis of breast cancer characteristics from the BCAC, CIMBA, and ENIGMA consortia. *Breast Cancer Res* **16**, 3419 (2014).
6. Bauer, K.R., Brown, M., Cress, R.D., Parise, C.A. & Caggiano, V. Descriptive analysis of estrogen receptor (ER)negative, progesterone receptor (PR)-negative, and HER2-negative invasive breast cancer, the so-called triple-negative phenotype - A population-based study from the California Cancer Registry. *Cancer* **109**, 1721-1728 (2007).
7. Zhang, J. *et al.* Comprehensive analysis of BRCA1 and BRCA2 germline mutations in a large cohort of 5931 Chinese women with breast cancer. *Breast Cancer Research and Treatment* **158**, 455-462 (2016).
8. Hakem, R. *et al.* The tumor suppressor gene *Brcal* is required for embryonic cellular proliferation in the mouse. *Cell* **85**, 1009-1023 (1996).
9. Shen, S.X. *et al.* A targeted disruption of the murine *Brcal* gene causes gamma-irradiation hypersensitivity and genetic instability. *Oncogene* **17**, 3115-24 (1998).

10. Liu, Q.H. *et al.* Chk1 is an essential kinase that is regulated by Atr and required for the G(2)/M DNA damage checkpoint. *Genes & Development* **14**, 1448-1459 (2000).
11. Takai, H. *et al.* Aberrant cell cycle checkpoint function and early embryonic death in Chk1(-/-) mice. *Genes & Development* **14**, 1439-1447 (2000).
12. O'Connell, M.J., Raleigh, J.M., Verkade, H.M. & Nurse, P. Chk1 is a wee1 kinase in the G2 DNA damage checkpoint inhibiting cdc2 by Y15 phosphorylation. *EMBO J* **16**, 545-54 (1997).
13. Lobry, C., Oh, P., Mansour, M.R., Look, A.T. & Aifantis, I. Notch signaling: switching an oncogene to a tumor suppressor. *Blood* **123**, 2451-2459 (2014).
14. Radtke, F. & Raj, K. The role of Notch in tumorigenesis: oncogene or tumour suppressor? *Nat Rev Cancer* **3**, 756-67 (2003).
15. Xie, X. *et al.* c-Jun N-terminal kinase promotes stem cell phenotype in triple-negative breast cancer through upregulation of Notch1 via activation of c-Jun. *Oncogene* **36**, 2599-2608 (2017).
16. Kim, D. *et al.* Notch1 in Tumor Microvascular Endothelial Cells and Tumoral miR-34a as Prognostic Markers in Locally Advanced Triple-Negative Breast Cancer. *J Breast Cancer* **22**, 562-578 (2019).

Reviewers' comments:

Reviewer #1 (Remarks to the Author):

The authors have already addressed most of the concerns and the manuscript is much improved. However, minor revision will be recommended due to the following reasons:

1. In line 208 -210, past tense should be used to describe the results.
2. In some western blots, the authors have performed semi-quantification to measure the band intensity but not in all the blots, especially in Fig 4.
3. According to the statistical section (M&M), the authors should mention their statistical tests used in the text or figure legends. However, I cannot find the test the authors used in results shown in Fig 3, Fig 4, Fig 7. What I can find are the * and NS in the figures. The authors should mention what kind of the test they used to determine statistical significance.

Reviewer #2 (Remarks to the Author):

In their revised version of the manuscript 'Blocking Non-canonical NOTCH1 Signalling Revives Mitotic Catastrophe Associate with BRCA1 Deficiency to Inhibit Triple Negative Breast Cancer', the authors have provided additional analyses and have adjusted the manuscript text.

The main point of critique is that there is a lack of mechanism on how ATR signaling is under control of Notch1. The authors indicate that this is a difficult experiment to do. However, without such data, the relation remains descriptive at best. The authors perform a number of experiments, but these only provide circumstantial evidence, and mechanism is not provided.

Also, the observation that in TNBC PDX models, cisplatin and ATR/Chk1 inhibitor works synergistically does not add anything new in the context of Notch1 signaling. also here causal relationships are lacking.

Specific points:

The data in R-Figure1 actually show that ATR depletion does prevent fibronectin accumulation (and hence EMT). In this case, shRNA#1 and #3 are the only shRNAs that work, and in these condition, FN levels are clearly not induced when compared to the other conditions. Conclusion is opposite to their conclusion. For the experiment with CHK1 shRNAs, there is such a high variety in background levels that it is impossible to use this data to state that Chk1 is not involved in EMT.

The observation that GSEA did not reach significance. I would advise this data to be included in the manuscript, not just the rebuttal letter. String analysis contains many interactions, including ones only based on being mentioned in the same abstract.

Concerning the shRNAs that target different regions of NOTCH1, I understand the rationale of the authors, although I don't think the method to read out differential knockdown is adequate. qPCR, or total levels of Notch1 protein is a better way to measure this. It remains unclear what the two bands are in the WB of 3e.

About the analysis of the DNA damage markers: images in the rebuttal letter are so small, that this is impossible to judge. The images suggest that 53BP1 is a pan nuclear protein, and relative intensity is used to read out DNA damage amounts. This is sub-standard in the field. Foci should be counted, and should be quantified. Also, better (higher magnification) images should be shown. Importantly, Figure R-16 suggests that the level of DNA damage is lowered upon Notch overexpression. Does this imply that cells are now able to repair their damage better? Of note: shBRCA1 gives very partial knockdown.

Replating assays (R-17): these data show that Notch1 induction (red line) prevents replating. In contrast, Brca1 depletion does not a major effect on proliferation. This is in contrast with their other data, and suggests that Notch prevents cell cycle progression. Also, Notch overexpression doesn't rescue the effects of Brca1 knockdown but makes it worse.

Reviewer #3 (Remarks to the Author):

The authors have performed an extraordinary amount of additional experiments to address the reviewers concern. I am satisfied with this revision.

Responses to Reviewers' comments

➤ Reviewer #1 (Remarks to the Author):

The authors have already addressed most of the concerns and the manuscript is much improved. However, minor revision will be recommended due to the following reasons:

Thanks a lot for your high evaluation. We have revised our manuscript according to your suggestions, which are detailed below.

1. In line 208 -210 (now line 213-215), past tense should be used to describe the results.

We have modified the sentence as following.

RNA-sequencing data also demonstrated that overexpression of Notch1 started from exons 25–27 (Fig. 3c), which is consistent with activated Notch1 protein expression analysis (Fig. 3d).

2. In some western blots, the authors have performed semi-quantification to measure the band intensity but not in all the blots, especially in Fig 4.

We have performed semi-quantification for all the blots. Kindly please check the manuscript.

3. According to the statistical section (M&M), the authors should mention their statistical tests used in the text or figure legends. However, I cannot find the test the authors used in results shown in Fig 3, Fig 4, Fig 7. What I can find are the * and NS in the figures. The authors should mention what kind of the test they used to determine statistical significance.

Thanks a lot for your kindness suggestion, we have described it in the Statistical analyses section (page 28, line 773-775), as well as the figure legends.

“The *t*-test was used to determine the significance of the difference between the different sets of data if there is not any specific indication.”

➤ Reviewer #2 (Remarks to the Author):

In their revised version of the manuscript ‘Blocking Non-canonical NOTCH1 Signalling Revives Mitotic Catastrophe Associate with BRCA1 Deficiency to Inhibit Triple Negative Breast Cancer’, the authors have provided additional analyses and have adjusted the manuscript text.

The main point of critique is that there is a lack of mechanism on how ATR signaling is under control of Notch1. The authors indicate that this is a difficult experiment to do. However, without such data, the relation remains descriptive at best. The authors perform a number of experiments, but these only provide circumstantial evidence, and mechanism is not provided.

Also, the observation that in TNBC PDX models, cisplatin and ATR/Chk1 inhibitor works synergistically does not add anything new in the context of Notch1 signaling. also here causal relationships are lacking.

Thank you very much for your comments, which are very helpful for our study. We have revised our manuscript according to your suggestions as following.

Specific points:

The data in R-Figure1 actually show that ATR depletion does prevent fibronectin accumulation (and hence EMT). In this case, shRNA#1 and #3 are the only shRNAs that work, and in these conditions, FN levels are clearly not induced when compared to the other conditions. Conclusion is opposite to their conclusion. For the experiment with CHK1 shRNAs, there is such a high variety in background levels that it is impossible to use this data to state that Chk1 is not involved in EMT.

The purpose of R-Figure 1 (now R2-Figure 1) is to study if the deletion of ATR or CHK1 could affect fibronectin accumulation. For ATR, shRNA1 and shRNA3 could knockdown ATR and shRNA2 could not (we have now put the knockdown value under the band). Our data indicated that in all cases, including shRNA1 and shRNA3, fibronectin levels were induced by Dox. For an easier illustration, we have normalized the value by setting “1” for 0 hour. The data revealed similar induction in all 4 cells. At 10 hr, the inductions were 1.96 in control, 2.12 in shRNA1, 2.36 in shRNA2, and 1.64 in sgRNA3 cells, respectively. Whereas at 20 hr, the inductions were 1.65 in control, 1.65 in shRNA1, 1.60 in shRNA2, and 1.41 in sgRNA3 cells, respectively. The data is consistent to the conclusion, i.e., the induction of fibronectin is independent of ATR.

R2-Figure 1. Western blot analysis of expression of Fibronectin during a time course of ICN1 induction upon knockdown of ATR and CHK1.

For CHK1, the “high variety in background levels” is primarily due to the fact that growth of cells is very sensitive to levels of CHK1. Our earlier study showed that CHK1^{-/-} (homozygous knockout) cells could not survive, whereas CHK1^{+/-} (heterozygous knockout) cells, although could survive, grow slower and display some abnormalities (Ref: Fishler, T., et al., Genetic instability and mammary tumor formation in mice carrying mammary-specific disruption of Chk1 and p53. *Oncogene*, 2010. 29(28): p. 4007-17). We believe this is the reason why shRNA-CHK1 could not reduce CHK1 levels more than 50%. Analysing these cells, our data indicated that the relative induction of fibronectin in these cells is similar to the control cells.

We noticed that in CHK1-shRNA2 cells, the basal level of fibronectin is increased. But since in CHK1-shRNA1 and CHK1-shRNA3 cells, reduced CHK1 did not increase fibronectin basal level, we do not believe CHK1 could negatively regulate fibronectin, and believed it might be a non-specific effect. Nonetheless, our experiments showed that in all 4 samples, fibronectin showed similar levels of induction after Dox administration.

The observation that GSEA did not reach significance. I would advise this data to be included in the manuscript, not just the rebuttal letter. String analysis contains many interactions, including ones only based on being mentioned in the same abstract.

Thanks a lot for your kindness suggestion, we have added it into supplementary tables (Supplementary Table 7).

Concerning the shRNAs that target different regions of NOTCH1, I understand the rationale of the authors, although I don't think the method to read out differential knockdown is adequate. qPCR, or total levels of Notch1 protein is a better way to measure this. It remains unclear what the two bands are in the WB of 3e.

As suggested, we conducted the Western blotting for full-length of Notch1 (upper panel) and qPCR (lower panels) to provide additional support. In the Notch1 driven tumor, as some of the ICN1 (activated Notch1) is driven by SB transposon to specifically overexpress the inner cell domain of Notch1, the shRNA-N (target to N-terminal portion) is not predicted to affect SB driven ICN1, but only deplete the endogenous full-length Notch1 and its activated Notch1. Our data indicated it indeed the case (see the first two lanes). Both the full-length Notch1 and ICN1 is predicted to be disrupted by shRNA-ICN1, and our data revealed it is indeed the case (the third lane).

In the Met driven tumor (non-Notch1 driven tumor), as the full-length of Notch1 is the only source of ICN1, which means all the ICN1 comes from full-length of Notch1 after ligands stimulation. And our analysis by Western blot (lanes 4-6) confirmed either shRNA-N or shRNA-ICN1 could reduce the expression level of both full-length of Notch1 and activated Notch1 (ICN1).

We have also conduct RT-QPCR analysis (lower panels) and the data is consistent with Western blot.

The two bands in the activated Notch1 western blotting may indicate endogenous activated Notch1 (upper band) and SB driven Notch1 (lower band).

R2-Figure 2. Western blot (Upper) and Q-PCR (Lower) analysis of Notch1-driven tumours (MK1370-3R) and non-Notch1-driven tumours (MK1097-5R) after shRNA knockdown.

About the analysis of the DNA damage markers: images in the rebuttal letter are so small, that this is impossible to judge. The images suggest that 53BP1 is a pan nuclear protein, and relative intensity is used to read out DNA damage amounts. This is sub-standard in the field. Foci should be counted, and should be quantified. Also, better (higher magnification) images should be shown.

Thank you very much for the suggestion, which really improve the quality of this manuscript. We have provided images with higher magnification and the foci quantification has been provided as Fig. 4c, d (also R2-Figure 3)

R2-Figure 3. IF staining and foci counting of γ -H2AX to indicate DNA damage at 48 hours after BRCA1 acute knockdown with or without ICN1 overexpression.

Importantly, Figure R-16 suggests that the level of DNA damage is lowered upon Notch overexpression. Does this imply that cells are now able to repair their damage better? Of note: shBRCA1 gives very partial knockdown.

Yes, we believe overexpression of ICN1 in BRCA1 mutant cells could facilitate DNA damage repair. Specifically, BRCA1 deficiency significantly impairs DNA damage repair (but not

completely shut down the repair ability, and some DNA damage might still be slowly repaired by some other machineries). Overexpression of ICN1 activates ATR-Chk1 signalling pathway, which slows down S phase progression and G2/M phase transition, and enables more time for BRCA1 mutant cells to repair DNA damage.

The shBRCA1 indeed gave a partial knockdown. This is primarily because BRCA1 is essential for viability of cells. We have tried to knockdown BRCA1 in many cell lines, and found acute knockdown of BRCA1 always initially causing lethality effect. BRCA1 knockdown cells could gradually grow and become better. However, when we check the expression of BRCA1, we found that the survived cells always contain a partial knockdown of BRCA1. We believe this is because loss of BRCA1, or acute knockdown of BRCA1, causes cellular lethality. The lower the BRCA1, the poorer the growth. Therefore, BRCA1 could only be partially knockdown in survival cells. In many BRCA1 partially knockdown cells, BRCA1 proliferation can be comparable with BRCA1 wild type cells. These cells might contain about 30-40% of BRCA1 compared with wild type cells. We believe that if their BRCA1 levels get lower, they will not survive.

Replating assays (R-17): these data show that Notch1 induction (red line) prevents replating. In contrast, Brca1 depletion does not have a major effect on proliferation. This is in contrast with their other data, and suggests that Notch prevents cell cycle progression. Also, Notch overexpression doesn't rescue the effects of Brca1 knockdown but makes it worse.

There might be a misunderstanding here primarily because we did not explain our data clearly enough. We are so sorry. We now would like to explain it in more details. The following 3 points we would like to explain first:

1) Overexpression of ICN1 could suppress the lethal effect caused by acute knockdown of BRCA1. For the rescue effect assay, we used the parental cells, which is naïve for the BRCA1 knockdown, therefore the knockdown of BRCA1 will cause cell death. In this case, when we overexpress the ICN1, the shBRCA1-cells grow better. (please compare the right upper and lower panels in R2-Figure 4 (This data is shown in Figure 4a, b).

R2-Figure 4. Overexpression of ICN1 suppressed cell death caused by Brca1 acute knockdown in MCF10A naïve cells.

2) High level expression of ICN1 could suppress cell proliferation in regardless of their genotype (i.e. both BRCA1 wild type and mutant, which could also be seen by comparing the left upper and lower panels in R2-Figure 4). We have conducted a dose response using 3 different concentration of DOX to induce NOTCH1 and found the higher the dose for induction of NOTCH1, the stronger is the suppression effect (R2-Figure 5).

R2-Figure 5. Re-plating assay for cell growth rate under different concentration of Dox induced ICN1 overexpression in MCF10A parental cell and BRCA1 knockdown adapted cells. Cell number was measured every three days, then re-plating the cells. Three repeats were used to determine the SD.

3) shBRCA1-cells gradually recover from poor growth and some of them can have similar growth rate compared with the parental wild type cells. However, these cells contain partial BRCA1 knockdown, usually cannot go less than 30-40% of wild type level as shown in R2-Figure 6 (also Figure 4l in the manuscript). We may call these cells sh-BRCA1 adapted cells. When overexpression of ICN1 in this type of shBRCA1-adapted cells, the effect is reduction of proliferation (rather than rescue lethality, as they are shBRCA1-adapted cells, which grow like wild type cells). In both BRCA1 knockdown and wild type cells, ICN1 could reduce proliferation because of the activation of ATR-CHK1 signaling to slow down the S/G2 and G2/M progression.

R2-Figure 6. Western blotting analysis demonstrate the knockdown level of BRCA1 in MCF10A cells.

With these 3 points, we would like to explain the replating assay:

Our replating assay was used to explain the data shown in Supplementary Figure 5h, that ICN1 overexpression reduces cell growth. In the replating assay, we used the shBRCA1-adapted cells. As explained earlier, the growth of these cells is comparable with wild type control cells, this is the reason why “Brca1 depletion does not have a major effect on proliferation”. In our earlier data, we used high concentration of Dox (1.5 $\mu\text{g/ml}$), which yielded much stronger inhibition effect on cell proliferation. After conducting a dose response, we now use a moderate concentration of Dox, 1 $\mu\text{g/ml}$, to do the experiment. The data showed that induction of NOTCH1 slows down cell proliferation in both control and shBRCA1-adapted cells (R2-Figure 7; Supplementary Figure 5i). The reduced proliferation of BRCA1-mutant cells enable them more time to repair DNA damage, of course, the long-term effect is the acceleration of tumorigenesis in these cells.

R2-Figure 7. Growth rate of MCF10A cells under ICN1 overexpression and/or Brca1 knockdown. Cell number was measured every three days, then re-plating the cells. Three repeats were used to determine the SD.

This data is placed in sFigure 5i. We have specifically indicated the cells used were shBRCA1 adapted cells in the text to avoid the misleading. Thanks so much for your kindness comments.

➤ **Reviewer #3 (Remarks to the Author):**

The authors have performed an extraordinary amount of additional experiments to address the reviewers concern. I am satisfied with this revision.

Thank you very much for the constructive suggestions and support for this work.

REVIEWERS' COMMENTS:

Reviewer #2 (Remarks to the Author):

I appreciate the additional work that the authors have done, although my main concern (lack of clear mechanism) remains valid..

the red text in the revised manuscript needs extensive editing; there are grammar issues on almost every sentence (including the new title).

- **REVIEWERS' COMMENTS:**
- **Reviewer #2 (Remarks to the Author):**

I appreciate the additional work that the authors have done, although my main concern (lack of clear mechanism) remains valid.

the red text in the revised manuscript needs extensive editing; there are grammar issues on almost every sentence (including the new title).

We have carefully modified our manuscript and have also sent it out to an editing service to correct any possible grammar errors.